# CYLON: ASYNCHRONOUS LINEAR ATTENTION

## ABSTRACT

Sequence models face stark tradeoffs between recall quality and memory efficiency. Recall – the ability to use information over long sequences – is critical for sequence modeling tasks ranging from information extraction to reasoning. Prior work has shown that in theory, *linear* attention models with sufficient recurrent state sizes can expand the Pareto frontier of the recall-memory tradeoff space beyond alternative architectures such as softmax attention and state space models. However, it is difficult to scale the linear attention state size due to hardware bottlenecks. I/O aware algorithms store the linear attention states in thread registers, however state sizes beyond $\approx 3$ megabytes exhaust register memory and trigger expensive register spills. In this work, we introduce CYLON, a hardware-aware strategy for partitioning linear attention's recurrent state across the registers of multiple GPU processors and asynchronously combining the partitions. When applying CYLON to popular architectures, such as Hedgehog and Mamba-2, we unlock $3\times$ higher throughput compared to prior linear attention algorithms for these architectures on both Hopper and Blackwell GPUs. Finally, CYLON makes large states available to model designers by unlocking sizes (e.g., $\geq 131$ MB) that are not achievable by the existing linear attention kernels.

## 1 INTRODUCTION

Sequence models face a fundamental tradeoff between recall quality—the ability to manage information over long sequences—and memory efficiency (Cho et al., 2014; Arora et al., 2024). While there has been significant exploration into architectures that require complex initializations and inductive biases over the past few years (von Oswald et al., 2025; Poli et al., 2023; Gu & Dao, 2023), the bitter lesson is that a model's memory consumption is the primary factor governing its quality (Arora et al., 2024). Prior work shows that using linear attention (Katharopoulos et al., 2020) with large recurrent state sizes expands the recall-memory frontier beyond prior architectures such as state space models (Gu et al., 2021) and softmax attention (Vaswani et al., 2017) (Figure 1).

Ideally, model designers could flexibly traverse the recall–efficiency tradeoff rather than being locked to one end of the spectrum. Applications land at different points on the tradeoff space: for example, low-latency workloads often want to prioritize extreme efficiency over recall (Qwen, 2025; Cartesia, 2025; Gu et al., 2025; Liquid, 2025). Today's models are at two extremes: softmax attention, which offers high recall but consumes hundreds of GBs of memory (Touvron et al., 2023), or linear attention, which only uses tiny recurrent states (2MB for Qwen3-Next, 2.6MB for Mamba2-2.7B) and sacrifices recall. We ask: *why is linear attention constrained to small state sizes and are there alternative sequence mixers that can use larger state sizes with improved efficiency?*

To understand why today's linear attention models use small state sizes, we start by profiling a suite of representative and popular linear attention kernels (Yang & Zhang, 2024). Linear attention replaces softmax attention with a simple dot product over query and key vectors. While softmax attention scales quadratically with sequence length, linear attention admits a sub-quadratic *chunk-wise parallel* algorithm (Hua et al., 2022): sequences are divided into chunks (e.g., of 16 tokens), and the output is computed using recurrent computations across chunks and parallel computations within chunks. The fastest kernels persistently store the recurrent state in thread registers, the fastest segment of GPU memory (Spector et al., 2025). However, register files are limited to 256 KB per processor (NVIDIA, 2022), so states beyond a few megabytes tend to trigger register spills to the GPU's L1 cache (resulting in $\approx 4\times$ slower memory accesses and stalls of $\approx 40$ cycles). Our analysis reveals that the key challenge is not raw compute but the cost of moving and aggregating state across the GPU memory hierarchy within a thread block (group of collaborating threads).

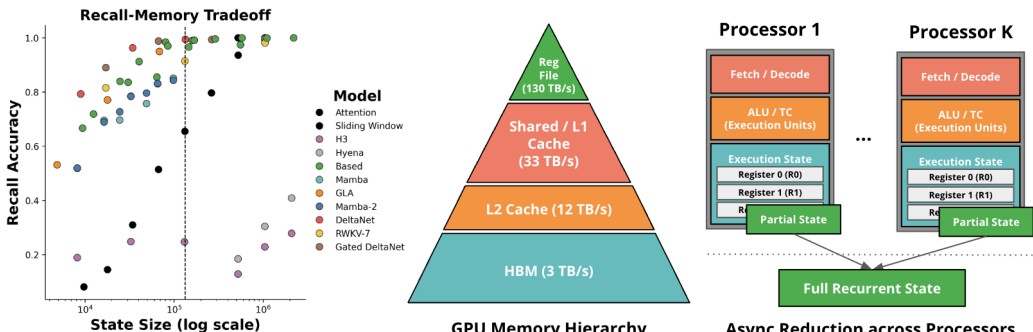

Figure 1: (**Left**) Despite substantial effort into architecture design, the bitter lesson is that a model's memory consumption is the primary factor governing its quality (Arora et al., 2023). Linear attention expands the frontier between quality and efficiency beyond prior architectures (SSMs, attention) (**Middle**) Hardware memory limits prevent foundation model designers from flexibly scaling the state size, however. (**Right**) CYLON addresses the bottleneck by partitioning the state across the registers of many processors and asynchronously combining them using modern GPU features.

We observe that current efficient architectures are missing a fundamental trend in AI accelerators that can help address the memory limits: accelerators are increasingly asynchronous machines. Our insight is, perhaps we can address the register bottleneck by partitioning large linear attention states across the fast memory of *multiple thread blocks*, and then asynchronously combine the partial results in the GPU's global memory. Unfortunately, old-school relaxed memory semantics atomic operations are bottlenecked by address generation, use of the generic (vs. async) proxy, and poor coalescing, which hurts memory throughput (Table 2). Excitingly, modern GPUs are prioritizing explicit hardware for bulk, pipelined asynchronous instructions such as Warp-group matrix multiply (WGMMA) and Tensor Memory Acceleration (TMA) on Hopper and Blackwell.

Inspired by the asynchrony of modern GPUs, we propose a hardware-aware modification to linear attention architectures called CYLON, designed to unlock new operating points in the recall-memory tradeoff space. We partition the recurrent state across $k$ streaming multiprocessors, each with a distinct featurization ($\phi_k$). Each SM maintains its partition in registers and processes the sequence independently with the chunkwise algorithm. A global asynchronous reduction is used to combine the partitioned states. CYLON is designed to maximize locality while enabling large effective state sizes, by using three classes of asynchronous operations:

- **Asynchronous multiplication:** Tensor cores execute the chunkwise matmuls using inputs in shared (or tensor) memory, reducing register pressure and allowing larger chunks per block.

- **Asynchronous I/O:** TMA streams each SM's partition between HBM and shared memory, with wave specialization to overlap data movement and computation (Bauer et al., 2011).

- **Asynchronous reductions:** Global reduction instructions (e.g., sum) aggregate the outputs of different SMs without explicit synchronization, avoiding HBM round-trips.

Overall, we demonstrate across two linear attention backbones (Hedgehog, Mamba-2) that CYLON achieves up to $3\times$ and $3.4\times$ greater throughput compared to widely used linear attention kernels on Hopper and Blackwell GPUs respectively, while matching the quality of linear attention on a suite of downstream recall-intensive and natural language understanding tasks, thereby expanding the Pareto frontier between quality and efficiency (Table 3). We additionally show that CYLON makes larger linear attention state sizes feasible compared to the baseline kernels (e.g., $\geq$131 MB) (Figure 3). In summary, our contributions include: (1) analysis of the bottlenecks of linear attention, (2) the CYLON architecture, and (3) an I/O-aware implementation of CYLON.

## 2 BACKGROUND

We provide background on GPU hardware in Section 2.1, background on efficient ML architectures in Section 2.2, and a profiling analysis of existing linear attention kernels in Section 2.3.

## 2.1 GPU Fundamentals

ML workloads are executed on GPUs as *kernels*, which are programs run by thousands of threads in parallel. In NVIDIA's terminology, GPUs consist of *streaming multiprocessors (SMs)*, each responsible for running the *thread blocks* that make up a kernel. At the hardware level, blocks are decomposed into *warps*–groups of 32 threads that execute in lockstep. A typical kernel loads data, performs computation, and writes results.

**Memory hierarchy** GPUs are designed with a layered memory hierarchy (Figure 1), each level trading off capacity for speed: (1) *Global memory (HBM):* Large but relatively slow, shared across all SMs (e.g., 80 GB, 3 TB/s bandwidth on NVIDIA H100), (2) *Shared memory (SRAM):* On-chip, fast, and scoped to a thread block within an SM (e.g., 256 KB, 33 TB/s). (3) *Registers:* Fast and small memory, private to individual threads (e.g., 65K registers per SM, 130 TB/s). Efficient kernel design carefully stages data through this hierarchy to minimize costly HBM accesses.

**Execution units** GPUs expose a diverse set of hardware execution units: (1) *Tensor Cores* are specialized units for dense matrix multiplications, delivering extraordinary throughput (approximately 1 PFLOP/s on H100). (2) *Arithmetic Logic Units and Special Function Units* are general-purpose compute units for scalar, vector, and transcendental operations, much slower in comparison (approximately 67 TFLOP/s on H100). (3) Processors have load/store units handle I/O. The performance gap between Tensor Cores and general ALU/SFU pipelines has widened across GPU generations, making tensor cores the critical path for modern ML workloads. Different warps can *occupy* different execution units, which helps avoid over-utilization of any one unit and improves throughput.

**Hardware asynchrony** Beyond occupancy, recent GPU generations introduce dedicated support for bulk, pipelined asynchronous execution. An asynchronous instruction can run in the background while threads perform other instructions. For example, Hopper and Blackwell architectures expose the Tensor Memory Accelerator (TMA) dedicated hardware for asynchrony:

- **TMA I/O:** TMA issues bulk data transfers between HBM and SRAM asynchronously. TMA performs address calculations, freeing registers and enabling producer-consumer specialization.
- **TMA reductions:** TMA can perform global reductions (e.g., sum, max, min) while SMs store to HBM, allowing reduction latency to be hidden behind ongoing computation.
- **Asynchronous MMA:** Instructions that allow Tensor Cores to consume operands directly from shared memory or tensor memory rather than registers. In practice, async MMAs are designed to work in tandem with TMA: TMA streams large tiles into SRAM asynchronously, and the tensor core consumes them efficiently, reducing register pressure and enabling larger tile sizes.

## 2.2 Efficient Architectures Tradeoff State Size and Quality

**Attention** Vaswani et al. (2017) define queries, keys, and values $\boldsymbol{q}, \boldsymbol{k}, \boldsymbol{v} \in \mathbb{R}^{N \times d}$ from input $\boldsymbol{u}$ via weight matrices $\boldsymbol{W}_q, \boldsymbol{W}_k, \boldsymbol{W}_v \in \mathbb{R}^{d \times d}$. The attention mechanism outputs

$$a_{n,i} = \frac{\exp\left(\boldsymbol{q}_n^\top \boldsymbol{k}_i / \sqrt{d}\right)}{\sum_{j=1}^n \exp\left(\boldsymbol{q}_n^\top \boldsymbol{k}_j / \sqrt{d}\right)}, \qquad \boldsymbol{y}_n = \sum_{i=1}^n a_{n,i} \boldsymbol{v}_i, \tag{1}$$

where $N$ is the sequence length and $d$ is the head dimension. Since each query $\boldsymbol{q}_n$ must interact with all prior keys $\boldsymbol{k}_i$, the overall runtime is quadratic in sequence length, $\mathcal{O}(N^2 d)$, while the memory cost during inference per token generation is $\mathcal{O}(Nd)$ due to the KV-cache.

**Linear attention** Katharopoulos et al. (2020) replace the softmax kernel with a feature map $\phi : \mathbb{R}^d \to \mathbb{R}^D$, yielding:

$$\hat{a}_{n,i} = \frac{\phi(\boldsymbol{q}_n)^\top \phi(\boldsymbol{k}_i)}{\sum_{j=1}^n \phi(\boldsymbol{q}_n)^\top \phi(\boldsymbol{k}_j)}, \qquad \hat{\boldsymbol{y}}_n = \sum_{i=1}^n \hat{a}_{n,i} \boldsymbol{v}_i. \tag{2}$$

Unlike softmax attention, the key–value terms can be pre-aggregated:

$$\hat{\boldsymbol{y}}_n = \frac{\phi(\boldsymbol{q}_n)^\top \left( \sum_{i=1}^n \phi(\boldsymbol{k}_i) \boldsymbol{v}_i^\top \right)}{\phi(\boldsymbol{q}_n)^\top \sum_{i=1}^n \phi(\boldsymbol{k}_i)}. \tag{3}$$

This leads to a recurrent update rule:

$$\hat{\boldsymbol{y}}_n = \frac{\phi(\boldsymbol{q}_n)^\top \boldsymbol{s}_n}{\phi(\boldsymbol{q}_n)^\top \boldsymbol{z}_n}, \qquad \boldsymbol{s}_n = \boldsymbol{s}_{n-1} + \phi(\boldsymbol{k}_n)\boldsymbol{v}_n^\top, \qquad \boldsymbol{z}_n = \boldsymbol{z}_{n-1} + \phi(\boldsymbol{k}_n), \qquad (4)$$

where $\boldsymbol{s}_n \in \mathbb{R}^{d \times D}$ is the recurrent state and $\boldsymbol{z}_n \in \mathbb{R}^D$ is a normalization accumulator. Recent work replaces $\boldsymbol{z}_n$ with layer or RMS normalization (Sun et al., 2023; Yang et al., 2023). Linear attention runs in $\mathcal{O}(NdD)$ time during training and $\mathcal{O}(Dd)$ time per recurrent step during inference, with constant memory $\mathcal{O}(Dd)$ independent of sequence length.

**State size and recall** Given the high cost of softmax attention, recent work has focused on linear attention variants (*e.g.*, Mamba-2, gated linear attention, RetNet (Dao & Gu, 2024; Yang et al., 2023; Sun et al., 2023)). Current evidence suggests that linear attention models generally underperform in quality compared to full attention (Arora et al., 2024). Increasing the state size improves quality by allowing the model to retain more information in its working memory during inference, but also incurs higher compute and memory costs, reducing efficiency (Figure 1, Left). State size is especially important in *recall tasks*, or tasks that require a model to manage and use long-term information, a capability that underpins reasoning, code generation, and dialogue.

### 2.3 PROFILING LINEAR ATTENTION

We profile representative linear attention kernels using the widely used Flash Linear Attention and Mamba-2 repositories on NVIDIA H100 GPUs (Yang & Zhang, 2024; Dao & Gu, 2024). We tune the Triton kernels and collect profiles using NVIDIA Nsight Compute (NCU). Experiments are run at sequence length 4096, batch size 8, and model dimension 2048 across different state sizes. We measure tensor core utilization, HBM throughput, and long scoreboard stalls (which correlate with stalls on HBM/L2 or register spills). Representative results are reported in Table 1 for models with 32 heads (small state size) and 1 head (large state size).

Linear attention admits three distinct factorizations, each with different hardware tradeoffs:

1. **Parallel:** Parallelizes over the sequence, multiplying $\mathbf{Q}\mathbf{K}^\top$ first (as in softmax attention, Eq. 1). Compute cost scales quadratically, $\mathcal{O}(N^2 d)$, making this form prohibitive for long sequences.
2. **Recurrent:** Pre-aggregates keys and values (Eq. 4) into a recurrent update rule. This minimizes FLOPs, $\mathcal{O}(NdD)$, but does not use tensor cores, limiting throughput.
3. **Chunkwise:** Interpolates between parallel and recurrent forms. Within chunks of size $C$, the parallel form uses tensor cores; across $\frac{N}{C}$ chunks, the recurrent form avoids quadratic compute.

Table 1 highlights these differences. The parallel kernel shows high tensor core utilization, but only due to wasted quadratic compute, resulting in low memory bandwidth and poor runtime. The recurrent kernel avoids quadratic compute but does not use tensor cores, yielding the slowest runtime. The chunkwise kernel is most efficient, but becomes register-bound at large state sizes, where recurrent state spills to L1 ($\sim$40 cycles, $\sim$4$\times$ slower). In contrast, the parallel kernel is unaffected by state size since it recomputes rather than persisting state in registers. We address the challenge of scaling to large state sizes by proposing CYLON in Section 3.

| Impl. | Duration (ms) | Utilizations (%) | | HBM | |
| --- | --- | --- | --- | --- | --- |
| | | Tensor core | | GB/s | Stalls (Cycles) |
| Parallel LA ($H = 32$) | 2.5 | 36.2 | | 157 | 1.5 |
| Parallel LA ($H = 1$) | 13.8 | 35.0 | | 129 | 1.5 |
| Recurrent LA ($H = 32$) | 6.0 | 0.0 | | 88 | 8.2 |
| Recurrent LA ($H = 1$) | 87.5 | 0.0 | | 57 | 5.5 |
| Chunkwise LA ($H = 32$) | 0.6 | 13.3 | | 1400 | 5.9 |
| Chunkwise LA ($H = 1$) | 4.8 | 24.7 | | 1620 | 7.1 |

Table 1: NCU profiles for linear attention kernels that use different factorizations.

## 3 CYLON: ASYNCHRONOUS LINEAR ATTENTION

Our objective is to enable large state sizes in linear attention while retaining hardware efficiency. Existing kernels struggle because recurrent states exceeding a few MBs cannot fit in registers, leading to register spills, shared memory contention, and global synchronization overhead (Section 2.3).

CYLON addresses this by partitioning the recurrent state across multiple SMs, ensuring each partition remains fully register-resident and then combining them asynchronously.

## 3.1 METHODOLOGY

Linear attention can be expressed using a single feature map function $\phi$, which transforms queries and keys before aggregation:

$$\boldsymbol{y}_n = \phi(\boldsymbol{q}_n)^\top \left( \sum_{i=1}^n \phi(\boldsymbol{k}_i)\boldsymbol{v}_i^\top \right). \tag{5}$$

CYLON generalizes this formulation by *partitioning* the computation across $m$ streaming multiprocessors (SMs). Each SM is assigned a distinct feature map $\phi_k$, producing independent partial states that are later aggregated:

$$\boldsymbol{y}_n = \sum_{k=1}^m \phi_k(\boldsymbol{q}_n)^\top \left( \sum_{i=1}^n \phi_k(\boldsymbol{k}_i)\boldsymbol{v}_i^\top \right). \tag{6}$$

As in standard linear attention, one may normalize Equation 6 with a normalization term:

$$\boldsymbol{y}_n = \frac{\sum_{k=1}^m \phi_k(\boldsymbol{q}_n)^\top \left( \sum_{i=1}^n \phi_k(\boldsymbol{k}_i)\boldsymbol{v}_i^\top \right)}{\sum_{k=1}^m \phi_k(\boldsymbol{q}_n)^\top \left( \sum_{i=1}^n \phi_k(\boldsymbol{k}_i) \right)}. \tag{7}$$

The linear attention state size is computed as $B(\frac{d^2}{H})$ for batch size $B$, model dimension $d$, and $H$ heads, where modulating $H$ changes the state size. For CYLON, the state size is computed as $Bm(\frac{d^2}{H})$, where modulating $H$ or $m$ changes the state size.

**Feature map functions.** Standard linear attention employs a *single* feature map $\phi(\cdot)$. In contrast, CYLON reuses the same $\boldsymbol{q}, \boldsymbol{k}, \boldsymbol{v}$ across SMs, but applies *distinct* $\phi_k(\cdot)$ functions. Simply duplicating the same map across SMs would yield redundant work; diversity among $\phi_k$ ensures that each SM contributes unique information. For quality, we implement $\phi_k$ as a learned MLP following Zhang et al. (2024). In practice, this reduces to a single linear projection followed by a softmax over the feature dimension:

$$\phi(x) = \frac{\exp(Ax + b)}{\sum_i \exp\left( (Ax + b)_i \right)}, \tag{8}$$

where $A$ and $b$ are learned weights.[1]

**Applicable Settings.** CYLON is a partitioning strategy which can be broadly applied to linear attention architectures. Linear attention variants such as Mamba, a state space model (SSM), can be CYLON-partitioned to more efficiently support large state sizes. To understand the impacts of CYLON, we compare baseline architectures to their CYLON-partitioned versions.

## 3.2 ANALYSIS

This section connects CYLON to existing architectures, linear attention (Katharopoulos et al., 2020) and softmax attention (Bahdanau et al., 2016).

**Linear attention.** A natural question is: *does splitting the recurrent state into disjoint partitions reduce the expressivity of linear attention?* We next investigate whether the partitioned representation, represented using feature maps $\phi_1, \ldots, \phi_m$, can recover the same computations as a single unified feature map, which we denote by $\Phi$.

**Theorem 1 (Unbounded Equivalence).** *Let $d, m, D$ be non-zero integers. Given a unified feature map $\Phi : \mathbb{R}^d \to \mathbb{R}^{mD}$, there exists a collection of smaller maps $\phi_1, \ldots, \phi_m : \mathbb{R}^d \to \mathbb{R}^D$ such that:*

$$\frac{\Phi(q_i)^\top \sum_{j \leq i} \left( \Phi(k_j)v_j^\top \right)}{\Phi(q_i)^\top \sum_{j \leq i} \Phi(k_j)} = \frac{\sum_{k=1}^m \phi_k(q_i)^\top \sum_{j \leq i} \left( \phi_k(k_j)v_j^\top \right)}{\sum_{k=1}^m \phi_k(q_i)^\top \sum_{j \leq i} \phi_k(k_j)}. \tag{9}$$

---

[1] In linear attention variants, such as Mamba-2, explicit feature maps conflict with the existing initialization (Dao & Gu, 2024). For these settings, we instead discard $\phi(\cdot)$ entirely and assign distinct projections of $\boldsymbol{q}$ and $\boldsymbol{k}$ across SMs, which remains efficient due to Mamba-2's Multi Value Attention design (see Appendix B.3).

A unified feature map can be decomposed into disjoint partitions without loss of expressivity.

**Theorem 2 (Elementwise Nonlinear Maps).** *Suppose $\Phi(x) = \sigma(Ax + b)$ with $A \in \mathbb{R}^{Dm \times d}$, $b \in \mathbb{R}^{Dm}$, and $\sigma$ an elementwise activation function. Let $C(\cdot)$ denote the number of arithmetic operations required by a function. Then there exist $\phi_1, \ldots, \phi_m$ satisfying Equation 9 such that $C(\Phi) = \sum_{k=1}^{m} C(\phi_k)$. In other words, the decomposition introduces* no additional computational cost *across all feature maps compared to the single, unified map.*

Together, these results establish that partitioning the state across SMs preserves the expressivity of linear attention. In practice, CYLON can match the representational capacity of non-partitioned models while benefiting from improved hardware efficiency. This helps explain why our models achieve comparable perplexity to baselines despite using distributed state (Section 4).

**Softmax attention.** Recall that softmax attention kernels require quadratic compute and centralized work within a single SM, where kernels are typically bottlenecked on the special functional units that compute the $\exp$ (Spector et al., 2025). While the quadratic compute cannot be avoided (Alman & Song, 2023), CYLON suggests an alternate path to softmax attention: the exponential function can be approximated to within machine precision using low-degree polynomial expansions (e.g., Chebyshev or Taylor series) (Zhang et al., 2024; Keles et al., 2023; Arora et al., 2024). Each SM is assigned a subset of polynomial terms in the expansion of $\exp(x)$ and runs the chunkwise algorithm for the term. The partial results are then combined via asynchronous global reductions (TMA). This assigns

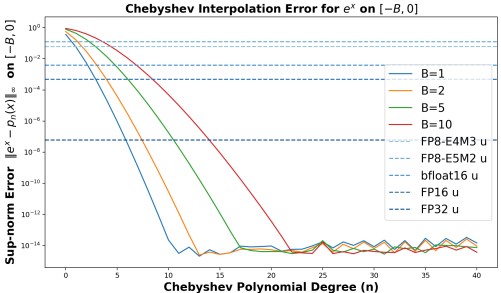

Figure 2: Sup-norm error $\|e^x - p_n(x)\|_\infty$ on $[-B, 0]$ for Chebyshev polynomials of degree $n$. Curves show different $B$ values; dashed horizontal lines mark half-ULP thresholds for FP8, bfloat16, FP16, and FP32.

the quadratic cost of softmax to a massively parallel grid of SMs. Motivating this feasibility, for common AI precision-levels (e.g., FP8), $\approx 3 - 5$ terms suffice for machine precision (Figure 2). This observation contextualizes CYLON's efficient expansion of feature dimension as an approach to approximating softmax and thus unifying softmax and linear attention in practice.

**Theorem 3 (Polynomial Expansion).** *Let $y = (Ax + b)^2$ for any $A \in \mathbb{R}^{Dm \times d}$, $b \in \mathbb{R}^{Dm}$, and $x \in \mathbb{R}^d$. There exists $\bar{A} \in \mathbb{R}^{Dm \times d^2 + d + 1}$, $\bar{b} \in \mathbb{R}^{Dm}$, and $\bar{x} \in \mathbb{R}^{d^2 + d + 1}$ such that $y = \bar{A}\bar{x} + \bar{b}$ is true.* Thus, the partitioning scheme we construct in Theorem 2 enables us to efficiently distribute the cost of a polynomial expansion across a GPU.

### 3.3 HARDWARE-AWARE ALGORITHMS

We make CYLON hardware-efficient by combining three systems principles: (1) *asynchronous execution*, (2) *work partitioning across SMs*, and (3) *register-locality of state*. Our design takes advantage of features such as Tensor Memory Accelerator (TMA) instructions and tensor cores (TC).

**CYLON kernel.** Given Equation 6, a straightforward implementation requires SMs to repeatedly communicate through high-bandwidth memory (HBM). Each SM must load the output tensor from HBM, add its local result, and write out the new tensor to HBM for the next SM. This introduces long-latency roundtrips and inter-SM synchronization overhead, producing up to a $16\times$ slowdown compared to our ultimate implementation.

Instead, CYLON partitions the recurrent state across SMs, keeps each partition fully resident in registers, and uses asynchronous reductions to combine results. Pseudocode for the core kernel is provided in Algorithm 2, where $\mathbf{local}_{KV}$ and $\mathbf{local}_K$ denote the cumulative register-resident states maintained by each SM. We also include the reference chunk-wise algorithm that is used in prior linear attention kernels for reference (Hua et al., 2022; Yang et al., 2023; Arora et al., 2024). The key operation is the reduction step, which aggregates results across SMs.

The kernel follows a memory-pipelined pattern, where warps overlap memory loads with compute by issuing asynchronous TMA loads to execute while performing computation on data resident in fast memory. Warps also overlap computation with inter-SM asynchronous reductions, eliminating

the need for global synchronization barriers. The kernel operates on chunk sizes of 64, chosen to align with native tensor core shapes. To maximize utilization, producers reallocate their registers to consumers once loading is complete, and consumers persist the recurrent state entirely in registers.

---

**Algorithm 1** Chunk-wise (Hua et al., 2022)

---

**Require:** Queries, keys, values $\boldsymbol{Q}, \boldsymbol{K}, \boldsymbol{V} \in \mathbb{R}^{N \times d}$ and feature map $\phi$
1: Initialize recurrent states $\mathbf{KV}, \mathbf{K} \leftarrow 0$ in registers
2: **for** $t = 0, \ldots, N/64$ **do**
3:      Load tile $Q_t, K_t, V_t$
4:      $X_t \leftarrow \phi(Q_t)\phi(K_t)^\top$
5:      Apply causal mask to $X_t$
6:      $O_t \leftarrow X_t V_t + \mathbf{KV}$
7:      $\mathbf{KV} \leftarrow \mathbf{KV} + \phi(K_t)^\top V_t$
8:      $\mathbf{K} \leftarrow \mathbf{K} + \phi(K_t)$

---

**Algorithm 2** CYLON async chunk-wise

---

**Require:** Queries, keys, values $\boldsymbol{Q}, \boldsymbol{K}, \boldsymbol{V} \in \mathbb{R}^{N \times d}$ and feature maps $\phi_1, \ldots, \phi_m$
1: Initialize local states $\mathbf{local}_{KV}, \mathbf{local}_K \leftarrow 0$ in registers
2: **for** $t = 0, \ldots, N/64$ **do**
3:      Load tile $Q_t, K_t, V_t$
4:      $X_t \leftarrow \phi(Q_t)\phi(K_t)^\top$
5:      Apply causal mask to $X_t$
6:      $O_t \leftarrow X_t V_t + \mathbf{local}_{KV}$
7:      **Asynchronous Reduce** $O_t$ across SMs
8:      $\mathbf{local}_{KV} \leftarrow \mathbf{local}_{KV} + \phi(K_t)^\top V_t$
9:      $\mathbf{local}_K \leftarrow \mathbf{local}_K + \phi(K_t)$

---

**Reduction Mechanisms.** A challenge in CYLON is efficiently aggregating partitioned states across SMs without becoming bottlenecked by memory synchronization. While traditional atomic operations (NVIDIA, 2020; AMD, 2025) enable concurrent updates through relaxed memory semantics, they suffer from address calculation overheads and poor memory coalescing, while also making it difficult to implement fine-grained pipelining (e.g., they use the generic, not async, proxy).

Modern GPUs expose explicit asynchronous reduction hardware through Tensor Memory Accelerator (TMA) instructions, which we leverage in CYLON. Unlike atomic operations that require address generation and use generic memory proxies, TMA reductions operate as bulk, pipelined operations - warps issue reductions (add, max, min) that execute asynchronously in dedicated hardware units while computation proceeds uninterrupted. Table 2 quantifies this advantage: TMA reductions achieve 8-11% higher throughput and 12-15% lower latency compared to atomic operations. The performance gap widens with larger feature maps (16 vs. 4). Compared to fully synchronous reductions, TMA achieves over 10x higher throughput and 2x lower latency. Moreover, synchronous reductions are limited by thread block cluster size on H100

| Mechanism | Feat. Maps | Latency (ms/iter) | Throughput (GB/s) | Notes |
|---|---|---|---|---|
| Sync reductions | 4 | 5.3 | 82.6 | Barrier-synchronized global |
| | 8 | 15.8 | 28.2 | memory access |
| Atomic ops | 4 | 2.5 | 856.9 | Serialized global memory access |
| | 16 | 9.3 | 845.4 | |
| Async reductions | 4 | 2.2 | 932.3 | Overlaps reduction with compute |
| | 16 | 7.9 | 940.9 | |

Table 2: Microbenchmark comparison of reduction mechanisms for 4 and 16 feature maps.

**Normalization relaxation.** Equation 7 requires summing numerators and denominators across all feature maps before dividing, which would force an expensive global memory roundtrip. SMs would need to asynchronously combine the numerators and denominators separately, before we load both results back into fast memory and perform the normalization. We balance quality and efficiency by introducing an empirically motivated learned scalar $e$ shared across feature maps:

$$\boldsymbol{y}_n = \sum_{k=1}^{m} \frac{\phi_k(\boldsymbol{q}_n)^\top \sum_{i=1}^{n} \phi_k(\boldsymbol{k}_i)\boldsymbol{v}_i^\top}{\phi_k(\boldsymbol{q}_n)^\top \sum_{i=1}^{n} \phi_k(\boldsymbol{k}_i) + e}. \tag{10}$$

The shared scalar $e$ provides a simple global correction term that compensates for drift across the magnitudes in different partitions.

| Architecture | Params/Tokens | State Sz. MB/layer | **LM** Slim Ppl. ↓ | **Info. Extraction** FDA Acc. ↑ | SWDE Acc. ↑ | **QA** SQUAD Acc. ↑ | **Long Ctx.** RULER Acc. ↑ | **Common** LM-Evals Acc. ↑ |
|---|---|---|---|---|---|---|---|---|
| Transformer | 1.3b/10b | 33.5 | 12.2 | 73.1 | 72.3 | 24.4 | 41.7 | 46.0 |
| Hedgehog m = 1 | 1.2b/10b | 1.1 | 12.4 | 22.3 | 40.3 | 33.3 | 21.2 | 45.7 |
| Hedgehog m = 2 | 1.2b/10b | 2.1 | 12.3 | 25.2 | 46.8 | **35.9** | **26.2** | **46.6** |
| Hedgehog | 1.2b/10b | 2.1 | 12.3 | **29.6** | **49.1** | 34.7 | 25.0 | 46.4 |
| Hedgehog m = 4 | 1.2b/10b | 4.2 | 12.8 | **25.7** | 43.9 | 32.4 | 22.1 | **47.0** |
| Hedgehog | 1.2b/10b | 4.2 | 12.7 | 22.9 | **44.7** | **32.7** | **23.4** | 45.8 |
| Mamba-2 m = 1 | 1.3b/10b | 2.1 | 12.3 | 16.6 | 42.6 | 32.1 | 20.4 | 46.5 |
| Mamba-2 m = 2 | 1.3b/10b | 4.2 | 12.1 | **23.8** | **45.2** | 34.6 | 26.4 | **46.6** |
| Mamba-2 | 1.3b/10b | 4.2 | 12.7 | 16.7 | 41.3 | 30.0 | **29.5** | 46.5 |
| Mamba-2 m = 4 | 1.3b/10b | 8.4 | 12.0 | **33.0** | **48.4** | 33.0 | **32.0** | 46.5 |
| Mamba-2 | 1.3b/10b | 8.4 | 12.3 | 28.2 | 47.8 | **34.5** | 30.1 | **46.7** |

Table 3: **Evaluation of pre-trained language models.** We train models of different state sizes (reported in MB per layer) using default linear attention and CYLON-partitioned linear attention. The variable $m$ denotes number of feature maps and the lack of an $m$ indicates a non-partitioned state. Models are trained using identical 10b tokens drawn from the Slim Pajama corpus. We report inference throughput on 4096 tokens of prefill with batch size 8. We report overall language modeling (LM) perplexity on a heldout test set of SlimPajama, and downstream scores on both recall-intensive (SWDE, FDA, SQUAD) and NLU tasks (LM-Evals). For each pair of models (CYLON and default linear attention) at the same state size, we **bold** the scores of the better performing model.

## 4 EXPERIMENTS

In this section, we present results to support the following claims:

1. **Language modeling overall.** We evaluate architectures in pretraining on the Slim Pajama dataset (Soboleva et al., 2023) and on standard language understanding (NLU) benchmarks. We find that CYLON-partitioned models match or exceed the quality of the non-partitioned (default) linear attention models across language modeling and NLU tasks.

2. **Language modeling recall.** We show that CYLON-partitioned state sizes preserve the recall quality of the default linear attention models. We show that scaling the state sizes correlates with recall quality improvements.

3. **Generation throughput.** Our CYLON kernel offers up to $3\times$ higher throughput compared to the popular SoTA linear attention kernels and also unlocks larger state sizes ($\geq 131$MB) than those supported by prior kernels. These results hold on both Hopper and Blackwell GPUs.

**Experimental setup.** We evaluate models at the 1.3B scale using 10B tokens of training on the Slim Pajama dataset (Soboleva et al., 2023) (tokenized with the GPT-2 BPE tokenizer (Radford et al., 2019)). We first pretrain and evaluate Transformer++ (rotary encodings (Su et al., 2023), gated linear units (Shazeer, 2020)) models and two representative variants of linear attention models, Hedgehog (Zhang et al., 2024) and Mamba-2 (Dao & Gu, 2024). We train each linear attention architecture across three state sizes. We then apply CYLON to Hedgehog and Mamba-2 using $m \in \{1, 2, 4\}$ state partitions, and compare the CYLON-model to the baseline model that uses the unified state. We run evaluations on downstream tasks using the LM Eval Harness (Gao et al., 2023).

### 4.1 LANGUAGE MODELING EVALUATIONS

We measure the perplexity of each pretrained model on a heldout test set of 10M tokens and provide the results in Table 3. We also evaluate these models on a suite of natural language understanding (NLU) tasks that are commonly used in architecture evaluations (SuperGLUE, ARC, PIQA, Wino-Grande, HellaSwag, LAMBADA). Quality on these tasks is reported in Table 3 ("LM-Evals"). We find that larger state sizes tend to result in lower perplexity scores, and that the CYLON models compete with the perplexity scores of the unified state models at each state size. We also observe that all model variants perform similarly on the NLU task suite. This is expected since the NLU tasks are relatively short context and thus do not implicate state size.

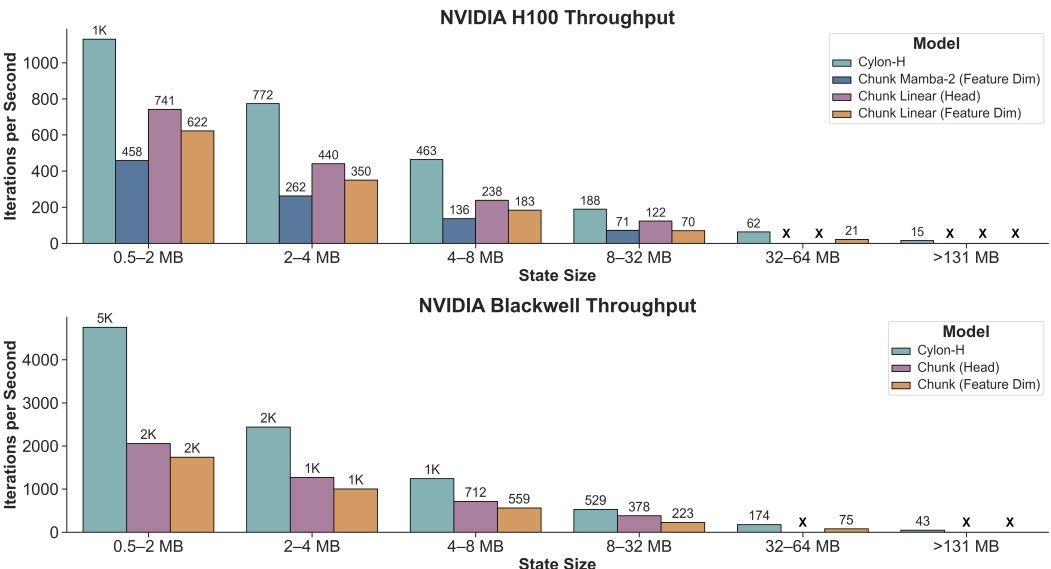

Figure 3: Performance across state sizes on NVIDIA Hopper and Blackwell GPUs. We report iterations per second for each method at different recurrent state sizes, grouped into size clusters. Missing measurements (denoted with "X") indicate configurations that exceed resource limits or are unsupported. CYLON offers up to $3\times$ higher throughput and unlocks large state sizes ($\geq 131$ MB).

## 4.2 RECALL EVALUATIONS

We evaluate our pre-trained models on a suite of downstream recall-intensive tasks proposed in prior work (Arora et al., 2024), including SWDE and FDA (Wu et al., 2021; Deng et al., 2022), two popular information extraction benchmarks, and SQUAD (Rajpurkar et al., 2018), a popular QA benchmark. Recall-intensive tasks require the model to remember information provided in the prompt context, which is a capability that benefits from using larger state sizes (Arora et al., 2024).

We observe that models with larger state sizes demonstrate improved recall quality, both for the models with CYLON partitioned and default recurrent states. The average recall score for CYLON models improves as 32.0 (m=1), 36.0 (m=2), and 34.0 (m=4) for Hedgehog, and improves as 30.4 (m=1), 34.5 (m=2) and 38.2 (m=4) for Mamba-2. We validate that the average scores of the CYLON models preserve the scores of the unified state models averaged across state sizes and tasks: across the 4 downstream task scores in Table 3, we see a 50% win-rate for CYLON models versus default Hedgehog models, and 75% win-rate for CYLON models versus default Mamba-2 models.

## 4.3 EFFICIENCY EVALUATIONS

We benchmark our CYLON kernel on both NVIDIA Hopper and Blackwell GPUs. In Figure 3, we compare to the widely used reference linear attentions (Flash linear attention (Yang & Zhang, 2024), Mamba-2 (Dao & Gu, 2024)). We benchmark kernels at sequence length 4096 tokens, model dimension 2048, and batch size 8 reporting the average of 100 iterations after 20 warmup iterations. We compute the model state sizes using the equations in Appendix A.

Figure 3 shows that CYLON achieves up to a $3\times$ speedup over chunkwise linear attention on both H100 and B200 GPUs. CYLON achieves up to a $3.4\times$ speedup over Mamba-2 on the H100 GPU. We observe that the reference kernels are brittle, failing to run (e.g., due to running out of memory) without error at the largest state sizes that we evaluate (e.g., $\geq 131MB$). Table 4 shows the impact of CYLON's speedups on end-to-end token throughput. This points to how CYLON unlocks a new operating point for AI: state size is critical to model quality, and CYLON allows model designers to use larger state sizes with significantly improved efficiency compared to prior methods.

| Architecture | Params | State Sz. MB/layer | Prefill Tok/Sec ↑ |
|---|---|---|---|
| Transformer | 1.3b | 33.5 | 9,656 |
| Hedgehog $m = 1$ | 1.2b | 1.1 | 9,997 |
| Hedgehog $m = 2$ | 1.2b | 2.1 | **9,670** |
| Hedgehog | 1.2b | 2.1 | 8,449 |
| Hedgehog $m = 4$ | 1.2b | 4.2 | **8,885** |
| Hedgehog | 1.2b | 4.2 | 7,559 |

Table 4: **End-to-end throughput comparison.** We report inference throughput on 4096 tokens of prefill with batch size 8 for Transformer and Hedgehog architectures with 1.3B and 1.2B parameters respectively.

## 5 CONCLUSION

This work tackles a long-standing bottleneck in linear attention: scaling recurrent states without collapsing throughput. Large states are essential for recall–a key skill for reasoning and long-context modeling–but linear attention suffers severe slowdowns as state scales. We propose CYLON, a hardware-aware architecture that partitions recurrent states across multiprocessors and combines them asynchronously. On Hopper and Blackwell GPUs, which embody the shift toward asynchronous execution, CYLON achieves up to $3\times$ higher throughput at small state sizes compared to widely used kernels. CYLON unlocks larger recurrent states ($\geq$131 MB) than supported by the baseline kernels. CYLON consistently preserves quality on downstream tasks, expanding the Pareto frontier of the recall–efficiency tradeoff. Overall, we show that CYLON makes large-state linear attention practical on modern GPUs, providing a path to efficient and scalable long-context models.

## 6 ETHICS STATEMENT

We adhere to the ICLR Code of Ethics and disclose our limited use of large language models (LLMs) in preparing this work. Specifically, we used LLMs for (i) high-level research ideation (e.g., brainstorming problem framings and organization of avenues to explore), and (ii) light assistance with initial proof sketches. All ideas were critically evaluated by the authors; all related-work leads were independently verified from the original sources before citation; the authors accept full responsibility for the final text. LLMs were not used to generate experiments, analyses, results, code, figures, or substantive technical content. No human subjects were involved, and we provided no confidential, proprietary, or personally identifiable data to LLM tools. We complied with data licenses and all data is open-source and readily available on Hugging Face. Our practices align with the Code's principles of research integrity, privacy, fairness, and legal compliance.

## 7 REPRODUCIBILITY STATEMENT

We provide all necessary pointers to replicate our results. The model and algorithmic details are specified in Section 3, with additional experimental details such as the exact hyperparameters used in Appendix A. We also list compute resources and the code we used for training in Appendix A. For theoretical results, we include complete proofs of the claims in Appendix C.

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

Our appendix is organized as follows:

1. Appendix A contains additional experimental details.
2. Appendix B contains additional experimental results.
3. Appendix C contains proofs for the theoretical analysis.

# A    EXPERIMENT DETAILS

We use A100 80GB NVIDIA GPUs to run all experiments. We use training infrastructure closely adapted from the FlashAttention code base: `https://github.com/Dao-AILab/flash-attention/tree/main` for all pretraining runs (Dao et al., 2022). Our pretraining data is pulled from the Slim Pajama corpus (Soboleva et al., 2023) and is tokenized using the GPT2BPETokenizer (Radford et al., 2019).[2] This section provides additional details for our model hyperparameters and configurations.

## A.1    MODEL CONFIGURATIONS

**Mamba-2 (Dao & Gu, 2024)**    We train using the implementation from the provided reference at `https://github.com/state-spaces/mamba`. We compute the state size for feature dimension $d_{\text{state}}$, model dimension $d$, value heads $H$, and expansion factor $E$ as:

$$\text{sizeof}(s_i) = E \times d_{\text{state}} \times d$$

Note that Mamba-2 uses multi value attention. We use the hyperparameters from Table 5.

Table 5: Mamba-2 Training Settings

| | |
|---|---|
| Optimizer | AdamW |
| Optimizer momentum | $\beta_1, \beta_2 = 0.9, 0.95$ |
| Optimizer eps | $1e-8$ |
| Precision | BFloat16 |
| Warmup | 1% |
| Learning rate decay | Cosine |
| Learning rate (min, base) | 8e-5, 8e-4 |
| Global batch size | 256 |
| Weight decay | 0.1 |
| Num Layers | 46 |
| Hidden Size | 2048 |
| RMSNorm | True |
| Norm Epsilon | $1e-5$ |
| Dt State | 16 |
| Dt (Min, Max) | (0.001, 0.1) |
| Dt Init. Strategy | Random |
| Dt Init. Floor | $1e-4$ |
| Dt Scale | 1.0 |
| Dt Softplus | True |
| Projection Expansion Factor | 2 |

**Hedgehog (Zhang et al., 2024)**    We use the implementation of Hedgehog from the provided reference at `https://github.com/HazyResearch/lolcats`. We compute the state size for feature dimension $d_{\text{feature}}$, model dimension $d$, and value heads $H$ as:

$$\text{sizeof}(s_i) = d_{\text{feature}} \times d$$

.

---

[2] `https://huggingface.co/datasets/DKYoon/SlimPajama-6B`

We train using the Based model backbone (Arora et al., 2024) from the provided reference at https://github.com/HazyResearch/based, which includes a hybrid of short convolution BaseConv (Arora et al., 2023), sliding window attention, and linear attention layers. This backbone provides a simple way to improve quality over pure linear attention. We use the hyperparameters from Table 6.

Table 6: Hedgehog Training Settings

| | |
|---|---|
| Optimizer | Adam |
| Optimizer momentum | $\beta_1, \beta_2 = 0.9, 0.95$ |
| Optimizer eps | $1e-8$ |
| Precision | BFloat16 |
| Warmup | 1% |
| Learning rate decay | Cosine |
| Learning rate (min, base) | 8e-5, 8e-4 |
| Global batch size | 256 |
| Weight decay | 0.1 |
| Num Layers | 26 |
| Hidden Size | 2048 |
| RMSNorm | True |
| Norm Epsilon | $1e-5$ |
| Dt State | 16 |
| Dt (Min, Max) | $(0.001, 0.1)$ |
| Dt Init. Strategy | Random |
| Dt Init. Floor | $1e-4$ |
| Dt Scale | 1.0 |
| Dt Softplus | True |
| Short Conv Filter Size | 4 |

## A.2  STATE SIZES OF OPEN-SOURCE MODELS

Throughout the paper, we discuss the state size of different popular linear attention models. Here we provide the basis for our computations:

| Model | Heads | $d_{q/k}$ | $d_v$ / $d_{\text{state}}$ | Elements | Size (Float32) |
|---|---|---|---|---|---|
| Qwen3-Next | 32 | 128 | 128 | $\approx 5.0 \times 10^5$ | $\sim 2.0$ MB |
| Jet-Nemotron | 16 | 128 | 256 | $\approx 5.0 \times 10^5$ | $\sim 2.0$ MB |
| GLA (FLA Hub) | 8 | 128 | 128 | $\approx 1.3 \times 10^5$ | $\sim 0.5$ MB |
| DeltaNet (Hub) | 20 | 128 | 128 | $\approx 3.3 \times 10^5$ | $\sim 1.3$ MB |
| Mamba2 2.7B | 80 | 64 | 128 | $\approx 6.5 \times 10^5$ | $\sim 2.6$ MB |

Table 7: Comparison of head configurations, dimensions, and state sizes.

For Mamba2 2.7B, we include the expansion factor of 2 in $d_{q/k}$.

# B ADDITIONAL RESULTS

Here we include additional experimental results.

## B.1 EXTENDED PROFILING FIGURES

We report additional profiling results in Table 8 to characterize the impact of asynchronous reductions.

| Kernel | Feature Maps | HBM Throughput (Gb/s) | Occupancy (%) | Tensor-Core Utilization (%) | Energy (J) |
|---|---|---|---|---|---|
| Linear Attention | 1 | 893.95 | 12.46 | 29.44 | 1.37 |
| Cylon | 1 | 834.47 | 12.36 | 24.04 | 1.41 |
| Cylon | 2 | 919.29 | 12.36 | 24.09 | 2.83 |
| Cylon | 4 | 926.48 | 12.36 | 24.13 | 5.63 |
| Cylon | 8 | 932.30 | 12.36 | 24.15 | 11.56 |
| Cylon | 16 | 940.88 | 12.36 | 24.16 | 23.74 |

Table 8: **Additional profiling metrics.** We compare our baseline Hedgehog linear attention kernel with CYLON. These kernels only differ by CYLON's use of asynchronous reduction operations in comparison to an asynchronous store.

## B.2 BENCHMARKS ON SXM H100

In Figure 3, we measure our H100 benchmarks on a H100 PCIe. Here we include H100 SXM benchmarks for completeness.

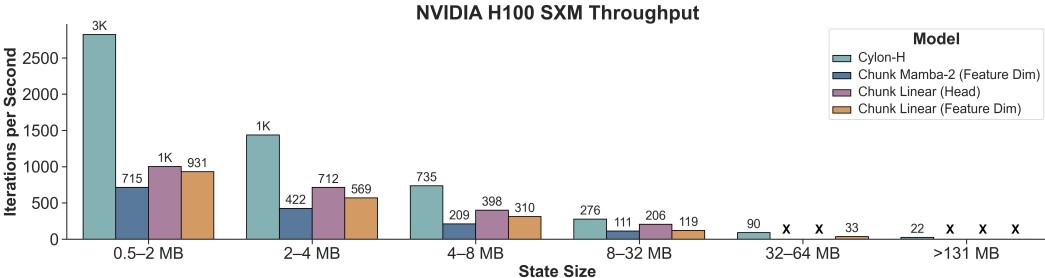

Figure 4: Performance across state sizes on NVIDIA Hopper (SXM) and Blackwell GPUs. We report iterations per second for each method at different recurrent state sizes, grouped into size clusters. Missing measurements (denoted with "X") indicate configurations that exceed resource limits or are unsupported.

## B.3 ARCHITECTURE ABLATIONS

**Learned Normalization Term.** In Section 3.3, we propose a learned scalar $e$ to compensate for our separate normalization computations across SMs. We provide ablation results for this choice in Table 9, demonstrating that including a normalization constant typically matches or improves accuracy on recall tasks FDA, SWDE, SQUAD, and RULER.

**Mamba2 Feature Activation.** We apply several non-linear element-wise maps to the BC and C tensors, as is ablated in the original Mamba-2 paper (Dao & Gu, 2024). In addition, we apply a learned Hedgehog feature map (Zhang et al., 2024). We find that our fixed non-linear maps do not impact the model's perplexity, confirming the results from the original Mamba-2 paper, but degrades its performance on recall evaluations by up to 7%. Moreover, we find that including a learned Hedgehog feature map degrades model performance by nearly 1 point of perplexity and recall evaluations by up to 7%.

| Architecture | $e$? | Params/Tokens | State Sz. MB/layer | **LM** Pile Ppl. ↓ | **Info. Extraction** FDA Acc. ↑ | SWDE Acc. ↑ | **QA** SQUAD Acc. ↑ | **Long Ctx.** RULER Acc. ↑ | **Common** LM-Evals Acc. ↑ |
|---|---|---|---|---|---|---|---|---|---|
| Hedgehog m = 1 | – | 1.2b/10b | 1.1 | 7.5 | 24.8 | 39.0 | 31.6 | 16.9 | 44.6 |
| Hedgehog m = 2 | Y | 1.2b/10b | 2.1 | **7.4** | 26.2 | **39.6** | **32.5** | 20.8 | 44.5 |
| Hedgehog m = 2 | N | 1.2b/10b | 2.1 | **7.4** | 29.6 | 39.5 | 31.6 | **32.0** | **44.8** |
| Hedgehog | – | 1.2b/10b | 2.1 | 7.5 | 25.9 | 38.2 | 32.3 | 20.1 | 44.7 |
| Hedgehog m = 4 | Y | 1.2b/10b | 4.2 | **7.4** | 26.4 | **45.4** | 31.6 | **29.0** | 44.2 |
| Hedgehog m = 4 | N | 1.2b/10b | 4.2 | 7.5 | 22.1 | 40.7 | **34.0** | 24.4 | **45.5** |
| Hedgehog | – | 1.2b/10b | 4.2 | 7.5 | 28.0 | 42.2 | 32.4 | 28.0 | 45.1 |

Table 9: **Ablation of shared normalization parameter.** We train models Hedgehog linear attention models with different state sizes and with/without the shared normalization parameter $e$, as proposed in Section 3.3. For each group of models (CYLON, with and without the shared normalization parameter, and default linear attention) at the same state size, we **bold** the scores of the better performing model.

## B.4 COST MODEL EXPERIMENTS

In Figure 5, we include a sweep of runtimes across state sizes (via head dimension ablation) and sequences lengths for the Flash Linear attention parallel, chunked, and fused chunked Triton kernels. We show that at large state sizes and sequence lengths, the chunked kernel is fastest.

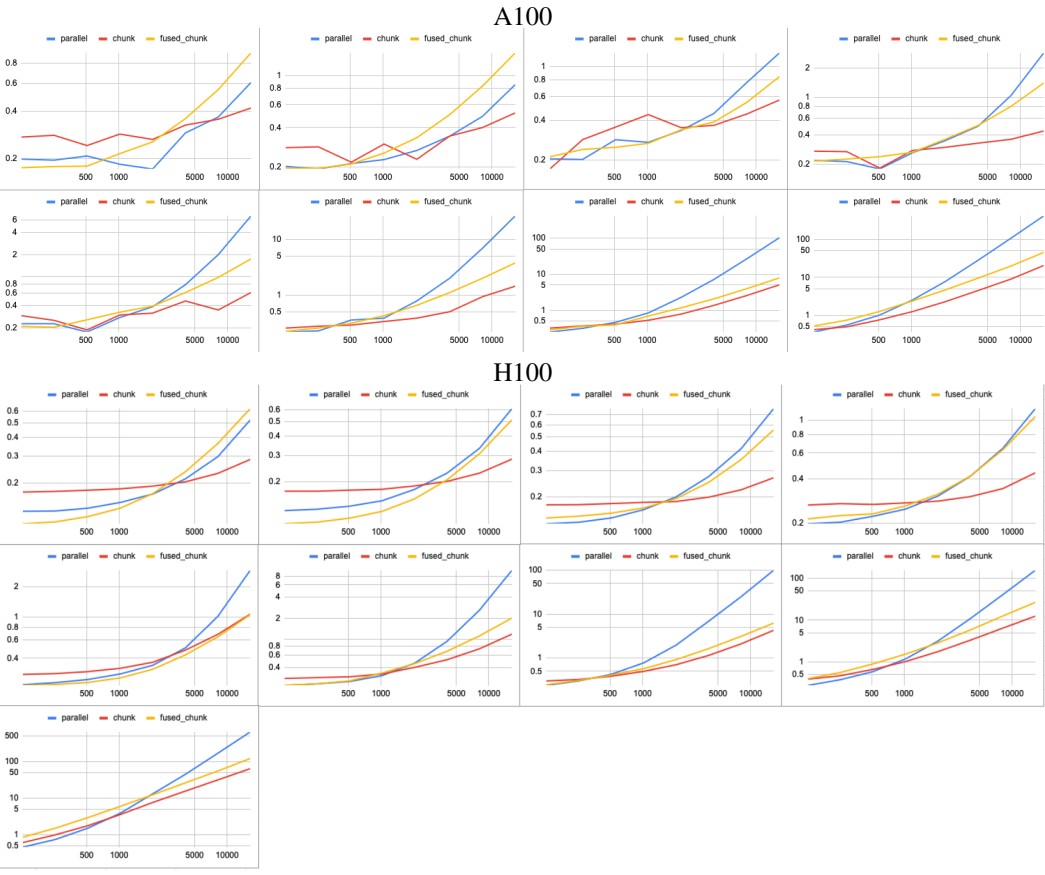

Figure 5: Sweep across prefill times. Each plot is set to a fixed head dimension. For both A100 and H100 sweeps, we start with a head dimension of 32 and increase by a factor of 2x in the plots left-to-right. For the A100, we end at a head dimension of 4096 in the lower right. For the H100, we end at a head dimension of 8192 in the lower left.

## B.5 EXTENDED MODELING EXPERIMENTS

The results in Table 3 are trained on Slim Pajama. To further validate our findings, we train a suite of models on The Pile. We find that the core trend holds: Cylon preserves the quality of linear attention models across evaluations. We also train several models on 50B tokens to show that our trends continue as we scale training duration. These results are presented in Tables 10 and 11 respectively.

| Architecture | Params/Tokens | State Sz. MB/layer | LM Pile Ppl. ↓ | Info. Extraction FDA Acc. ↑ | SWDE Acc. ↑ | QA SQUAD Acc. ↑ | Long Ctx. RULER Acc. ↑ | Common LM-Evals Acc. ↑ |
|---|---|---|---|---|---|---|---|---|
| Transformer | 1.3b/10b | 33.5 | **7.2** | **76.0** | **68.6** | 36.5 | **46.3** | **45.2** |
| Hedgehog m = 1 | 1.2b/10b | 1.1 | 7.5 | 24.8 | 39.0 | 31.6 | 16.9 | 44.6 |
| Hedgehog m = 2 | 1.2b/10b | 2.1 | **7.4** | **26.2** | **39.6** | **32.5** | **20.8** | 44.5 |
| Hedgehog | 1.2b/10b | 2.1 | 7.5 | 25.9 | 38.2 | 32.3 | 20.1 | **44.7** |
| Hedgehog m = 4 | 1.2b/10b | 4.2 | **7.4** | 26.4 | **45.4** | 31.6 | **29.0** | 44.2 |
| Hedgehog | 1.2b/10b | 4.2 | 7.5 | **28.0** | 42.2 | **32.4** | 28.0 | **45.1** |
| Mamba-2 m = 1 | 1.3b/10b | 2.1 | 7.4 | 19.4 | 38.2 | 30.6 | 22.6 | 44.4 |
| Mamba-2 m = 2 | 1.3b/10b | 4.2 | 7.4 | 20.2 | 41.8 | **33.2** | 30.2 | **45.0** |
| Mamba-2 | 1.3b/10b | 4.2 | **7.3** | **22.0** | **44.1** | 30.2 | **36.0** | **45.0** |
| Mamba-2 m = 4 | 1.3b/10b | 8.4 | 7.4 | **38.4** | **48.4** | 30.4 | 21.5 | **45.2** |
| Mamba-2 | 1.3b/10b | 8.4 | **7.4** | 31.3 | 45.5 | **32.7** | **29.9** | 44.8 |

Table 10: **Evaluation of pre-trained language models trained on The Pile.** We train models of different state sizes (reported in MB per layer) using default linear attention and CYLON-partitioned linear attention. The variable $m$ denotes number of feature maps and the lack of an $m$ indicates a non-partitioned state. Models are trained using identical 10b and 50b tokens drawn from The Pile corpus. We report overall language modeling (LM) perplexity on a heldout test set of The Pile, and downstream scores on both recall-intensive (SWDE, FDA, SQUAD, RULER) and NLU tasks (LM-Evals). For each pair of models (CYLON and default linear attention) at the same state size, we **bold** the scores of the better performing model.

| Architecture | Params/Tokens | State Sz. MB/layer | LM Pile Ppl. ↓ | Info. Extraction FDA Acc. ↑ | SWDE Acc. ↑ | QA SQUAD Acc. ↑ | Long Ctx. RULER Acc. ↑ | Common LM-Evals Acc. ↑ |
|---|---|---|---|---|---|---|---|---|
| Transformer | 1.3b/50b | 33.5 | 6.6 | 76.0 | 73.0 | 39.0 | 48.6 | 47.6 |
| Hedgehog m = 1 | 1.2b/50b | 0.6 | 6.8 | 25.6 | 45.9 | 33.5 | 38.3 | 47.7 |
| Hedgehog m = 4 | 1.2b/50b | 2.1 | **6.8** | **35.8** | 50.1 | **36.1** | **37.1** | **47.5** |
| Hedgehog | 1.2b/50b | 2.1 | **6.8** | 34.6 | **51.0** | 34.7 | 33.1 | 46.9 |

Table 11: **Evaluation of pre-trained language models trained on 50B tokens.** Similar to Tables 3 and 10, we train models of different state sizes using default linear attention and CYLON-partitioned linear attention. We train these models for 50B tokens each to validate our results at scale. We evaluate these models across a suite of tasks and, for each pair of models at the same state size, we **bold** the scores of the better performing model.

## B.6 TRAINING STABILITY

In Figure 6, we plot feature map gradient statistics to demonstrate training stability across models with multiple feature maps.

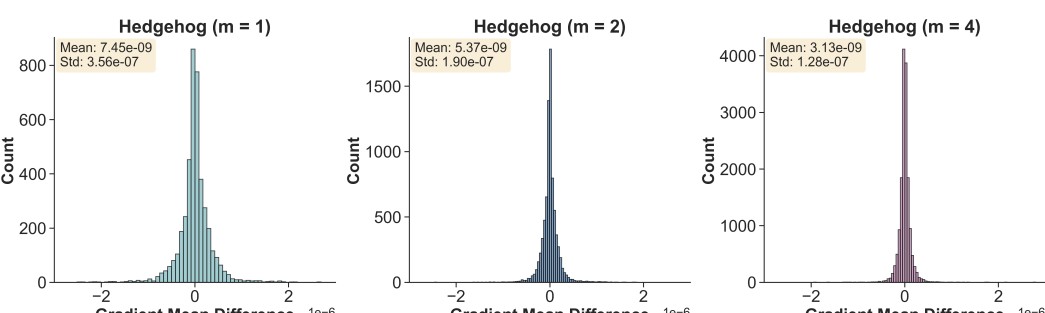

Figure 6: **Training Stability.** During training, we measure each parameter tensor's gradient mean, max, and min at a regular interval of 26 steps. We compute the inter-step differences in gradient mean across all feature maps' linear projections. We plot a distribution of these differences for three Hedgehog 1B parameter models trained on 10B tokens from The Pile.

## C  THEORETICAL ANALYSIS

A natural question is: *how does partitioning the state into disjoint components impact expressivity?* More formally, we study whether the behavior of linear attention with a single, non-partitioned feature map $\Phi$ can be reproduced with $m$ independent feature maps $\phi_1, \ldots, \phi_m$. In particular, we will assume a fixed $\Phi$ with some set of properties and then show that we can prove the existence of $\phi_1, \ldots, \phi_m$ which satisfies the following

$$\frac{\Phi(q_i)^\mathsf{T} \sum_j^i \left( \Phi(k_j) v_j^\mathsf{T} \right)}{\Phi(q_i)^\mathsf{T} \sum_j^i \phi(k_j)} = \frac{\sum_k^m \phi_k(q_i) \sum_j^i \left( \phi_k(k_j)^\mathsf{T} v_j \right)}{\sum_k^m \phi_k(q_i) \sum_j^i \phi_k(k_j)} \tag{11}$$

where $q_i, k_j, v_j \in \mathbb{R}^d$, $\Phi : \mathbb{R}^d \to \mathbb{R}^{mD}$, and $\phi_k : \mathbb{R}^d \to \mathbb{R}^D$ for all $i$ such that $1 \le i \le m$.

### C.1  UNBOUNDED EQUIVALENCE

Let us prove Equation 11 without making any assumptions regarding the structure of $\Phi$ or placing significant restrictions on $\phi_1, \ldots, \phi_m$.

**Proof:** First, let us construct $\phi_1, \ldots, \phi_m$. Let $\phi_i(x) = \Phi(x)[i \cdot D : (i+1) \cdot D]$ [3]. Given this construction, we know that $\Phi(x) = [\phi_1(x)^\mathsf{T}, \ldots, \phi_m(x)^\mathsf{T}]^\mathsf{T}$ is true. Thus, we can derive the following.

$$\frac{\phi(q_i)^\mathsf{T} \sum_j^n \left( \phi(k_j) v_j^\mathsf{T} \right)}{\phi(q_i)^\mathsf{T} \sum_j^n \phi(k_j)} = \frac{[\phi_1(q_i)^\mathsf{T}, ..., \phi_m(q_i)^\mathsf{T}] \sum_j^n \left( [\phi_1(k_j)^\mathsf{T}, ..., \phi_m(k_j)^\mathsf{T}]^\mathsf{T} v_j^\mathsf{T} \right)}{[\phi_1(q_i)^\mathsf{T}, ..., \phi_m(q_i)^\mathsf{T}] \sum_j^n [\phi_1(k_j)^\mathsf{T}, ..., \phi_m(k_j)^\mathsf{T}]^\mathsf{T}} \tag{12}$$

$$= \frac{\sum_j^n [\phi_1(q_i)^\mathsf{T}, ..., \phi_m(q_i)^\mathsf{T}][\phi_1(k_j)^\mathsf{T}, ..., \phi_m(k_j)^\mathsf{T}]^\mathsf{T} v_j}{\sum_j^n [\phi_1(q_i)^\mathsf{T}, ..., \phi_m(q_i)^\mathsf{T}][\phi_1(k_j)^\mathsf{T}, ..., \phi_m(k_j)^\mathsf{T}]^\mathsf{T}} \tag{13}$$

$$= \frac{\sum_j^n \sum_k^m (\phi_k(q_i)^T \phi_k(k_j)) v_j}{\sum_j^n \sum_k^m (\phi_k(q_i)^T \phi_k(k_j))} \tag{14}$$

$$= \frac{\sum_j^n \sum_k^m (\phi_k(q_i)^T \phi_k(k_j)) v_j}{\sum_j^n \sum_k^m (\phi_k(q_i)^T \phi_k(k_j))} \tag{15}$$

$$= \frac{\sum_k^m \phi_k(q_i)^T \sum_j^n \phi_k(k_j) v_j}{\sum_k^m \phi_k(q_i)^T \sum_j^n \phi_k(k_j)} \tag{16}$$

$$\tag{17}$$

Thus, we have demonstrated Equation 11 in a minimal setting. One may notice that this is unsatisfying, however, since $\phi_1, \ldots, \phi_m$ are unrestricted and thus enjoy unlimited computational resources. In the following sections, we further characterize our claims to be more meaningful in our applied setting, where we require $\phi_1, \ldots, \phi_m$ to be individually "cheaper" than $\Phi$.

### C.2  STRICTLY BOUNDED EQUIVALENCE

Let us introduce structure to $\Phi$. Pick some $A \in \mathbb{R}^{d \times Dm}$ and $b \in \mathbb{R}^{Dm}$ and let $\sigma$ be some non-linear function. We define the function $\Phi(x) = \sigma(Ax + b)$ for any $x \in \mathbb{R}^d$. In this section, we will prove Equation 11 assuming that $\sigma$ is an elementwise function [4]. More formally, for any $z \in \mathbb{R}^a$ for some arbitrary non-zero integer $a$, $\sigma(z)[i] = \sigma(z[i])$ is true for some index $1 \le i \le a$. Moreover, let $C(\cdot)$ denote the number of primitive operations required by a function. Primitive operation include arithmetic operations, exponentiation/logarithms, ReLU, and other elementwise operations. Let us place an additional restriction on $\phi_1, \ldots, \phi_m$ such that $C(\Phi) = \sum_{k=1}^m C(\phi_k)$ must be true.

---

[3]We borrow the $A[x : y]$ operator from Python, where it denotes a "slice" of tensor $A$ ranging from position $x$ to $y$ in its first dimension

[4]In practice, we set $\sigma$ to be some non-linearity such as the exponential or ReLU

**Proof:** Let us prove Equation 11 by constructing $\phi_1, \ldots, \phi_m$ such that $\Phi(x) = \left[\phi_1(x)^\mathsf{T}, \ldots, \phi_m(x)^\mathsf{T}\right]^\mathsf{T}$. If we can demonstrate that our construction satisfies this equality, then our proof is complete via our previous result.

We can produce this construction by decomposing $A$ and $b$ along their second and first dimensions, respectively, to produce $A_1, \ldots, A_m \in \mathbb{R}^{D \times d}$ and $b, \ldots, b \in \mathbb{R}^D$. Then, we define $\phi_i(x) = \sigma(A_i x + b_i)$ where $\sigma$ is the same elementwise operation from $\phi$. Now, we can easily derive the following. Recall that since $\sigma$ is element-wise, we can write $\sigma(z) = [\sigma(z_1), \ldots, \sigma(z_a)]$ if $z = [z_1, \ldots, z_a]$.

$$\phi(x) = \sigma(Ax + b) \tag{18}$$
$$= \sigma\left([A_1^\mathsf{T}, \ldots, A_m^\mathsf{T}]^+ [b_1^\mathsf{T}, \ldots, b_m^\mathsf{T}]^\mathsf{T}\right) \tag{19}$$
$$= \sigma\left([(A_1 x + b_1)^\mathsf{T}, \ldots, (A_m x + b_m)^\mathsf{T}]^\mathsf{T}\right) \tag{20}$$
$$= [\sigma(A_1 x + b_1)^\mathsf{T}, \ldots, \sigma(A_m x + b_m)^\mathsf{T}]^\mathsf{T} \tag{21}$$
$$= [\phi_1(x)^\mathsf{T}, \ldots, \phi_m(x)^\mathsf{T}]^\mathsf{T} \tag{22}$$

This proves our desired statement. From this, Equation 11 follows as a result of the previous proof.

Now, let us prove our second equality regarding total operations. We know that $C(\Phi) = dDm + (d-1)Dm + Dm + Dm = (2d+1)Dm$. The first two terms account for computing $Ax$, the third term accounts for adding bias, and the final term accounts for computing $\sigma$. Moreover, by similar logic we know $C(\phi_k) = dD + (d-1)D + D + D = (2d+1)D$ for all $1 \leq k \leq m$. Thus, we know that $\sum_{k=1}^{m} C(\phi_k) = \sum_{k=1}^{m} (2d+1)D = (2d+1)Dm$, which is equal to $C(\Phi) = (2d+1)Dm$, as required. Thus concludes our proof, in which we demonstrated a simple construction for partitioning $\phi$l, showed equivalence between our construction between our the partitioned and non-partitioned linear attention expressions, and proved that they use equal numbers of operations to compute.

### C.3 POLYNOMIAL EXPANSION.

Let $y = (Ax + b)^2$ for any $A \in \mathbb{R}^{Dm \times d}$, $b \in \mathbb{R}^{Dm}$, and $x \in \mathbb{R}^d$. There exists $\bar{A} \in \mathbb{R}^{Dm \times d^2+d+1}$, $\bar{b} \in \mathbb{R}^{Dm}$, and $\bar{x} \in \mathbb{R}^{d^2+d+1}$ such that $y = \bar{A}\bar{x} + \bar{b}$ is true.

**Proof:** Let us expand our expression as follows.

$$y_i = \left(\sum_j A_{ij} x_j + b_i\right)^2 \tag{23}$$
$$= \sum_j \sum_k A_{ij} A_{ik} x_k x_j + 2b_i \sum_j A_{ij} x_j + b_i^2 \tag{24}$$

From this, we can produce the following construction. Let "vec" below denote the "flattening" of a matrix to a vector.

$$\bar{A}_i = [A_{i1}A_{i1}, A_{i1}A_{i2}, \ldots, A_{i1}A_{id}, A_{i2}A_{i1}, \ldots, A_{id}A_{id} \mid 2b_i A_{i1}, 2b_i A_{i2}, \ldots, 2b_i A_{id} \mid 0] \tag{25}$$
$$\bar{b} = \left[b_1^2, b_2^2, \ldots, b_{Dm}^2\right]^\mathsf{T} \tag{26}$$
$$\bar{x} = [\text{vec}(xx^\mathsf{T})^\mathsf{T}, x^\mathsf{T}, 1]^\mathsf{T} \tag{27}$$

If we substitute our constructions into (24), we produce our desired equality $y = \bar{A}\bar{x} + \bar{b}$, thus proving that a $\bar{A}, \bar{b}, \bar{x}$ exist which satisfy our claim.

