# OpenReview forum: "Cylon: Asynchronous Linear Attention"
_ICLR.cc/2026/Conference — Submitted to ICLR 2026_

### Official Review · Reviewer_b6Yf · 2025-10-30

**Soundness:** 3
**Presentation:** 3
**Contribution:** 2
**Rating:** 4
**Confidence:** 4

**Summary:**

This paper addresses the fundamental tradeoff between recall quality and memory efficiency in sequence models. The authors identify that while linear attention models can theoretically outperform alternatives by using large recurrent states, they are practically limited by a hardware bottleneck: recurrent states larger than a few megabytes ($\approx3$ MB) exhaust fast GPU register memory, causing expensive "register spills" to the L1 cache.

To solve this, the authors propose CYLON, a hardware-aware linear attention architecture44. CYLON's core mechanism is to partition a large recurrent state across the registers of multiple streaming multiprocessors (SMs). Each SM processes its partition independently, and the partial results are then combined efficiently using modern asynchronous GPU operations, particularly the Tensor Memory Accelerator (TMA) for I/O and asynchronous global reductions.

**Strengths:**

1. Clear Problem Identification and Analysis: The paper does an excellent job of identifying a specific, critical, and practical bottleneck in modern sequence models. The analysis in Section 2.3, supported by Table 1, clearly profiles existing linear attention kernels and pinpoints register spills as the key limiting factor for scaling recurrent state size in efficient chunkwise algorithms.

2. Strong Empirical Performance: The performance improvements are a major strength. The paper demonstrates up to 3x higher throughput (iterations per second) on both H100 and Blackwell GPUs (Figure 3). This is a significant speedup that makes these models more practical.

3. Expanding the Pareto Frontier: Perhaps the most important contribution is that CYLON doesn't just speed up existing models; it enables new models. By unlocking state sizes ($\ge 131$ MB) that prior kernels could not support, the paper delivers on its promise of expanding the recall-efficiency trade-off space. This allows model designers to flexibly scale state size for better recall, which was previously not feasible

**Weaknesses:**

1. Normalization Relaxation (Eq. 10): A critical step for efficiency is the "normalization relaxation" proposed in Equation 1025. This avoids a global memory roundtrip 26by introducing a single "learned scalar $e$" 27instead of performing the full, correct normalization shown in Equation 728. While the results in Table 3 suggest quality is preserved29, this is a significant approximation. The paper lacks an ablation study to quantify the quality (and performance) impact of this specific relaxation. It is unclear if the quality parity is due to the partitioning being truly equivalent (as Theorems 1-2 suggest) or if this approximation happens to work well for the tested tasks.

2. Complexity of Partitioned Feature Maps: The CYLON architecture requires $m$ distinct feature maps ($\phi_k$) 30303030, which are implemented as learned MLPs 3131, to ensure diversity32. This seems to add implementation and potential training complexity compared to a standard linear attention model. The paper does not fully explore the parameter overhead or the impact on training dynamics (e.g., convergence, stability) resulting from this design choice.

3. Unsubstantiated "Path to Softmax": Section 3.2 also proposes an "alternate path to softmax attention" by approximating the $exp(x)$ function with polynomial expansions distributed across SMs 33. This is an interesting concept but is presented without any empirical validation. It feels disconnected from the paper's main contribution (linear attention) and leaves the reader wondering if this is a practical proposal or a purely theoretical aside.

**Questions:**

1. Could the authors provide an ablation study on the normalization relaxation in Equation 10? What is the quality and throughput impact of implementing the "correct" normalization (Equation 7) —which would require a global synchronization—compared to the proposed relaxation? This would help isolate the gains from partitioning from those from this approximation.

2. Regarding the proposed "alternate path to softmax attention": Have the authors benchmarked this approach? How does its practical throughput and, more importantly, its numerical stability (especially with low-precision formats like FP8) compare to standard, fused-kernel implementations of softmax attention like FlashAttention?

3. How sensitive is the model's quality and performance to the number of partitions, $m$? Table 3 shows results for $m \in \{1, 2, 4\}$, but the throughput benchmarks in Figure 3 are not broken down by $m$. How does throughput scale as $m$ increases, and is there a point where the overhead of the asynchronous reduction outweighs the benefit of smaller, register-resident partitions?

4. The paper focuses on inference throughput, specifically prefill. How does CYLON affect training performance? Does the partitioned architecture and the addition of $m$ distinct feature maps impact the training-time compute graph, memory footprint, or end-to-end training speed?

---

> ### Author Response · Authors · 2025-11-23
> **Response to Review**
>
> Thank you for the questions and suggestions! We have included additional results below and in our revised paper to address your comments. Your feedback has helped us further refine and strengthen our work.
>
> ## Normalization relaxation.
>
> Thank you for the question regarding the normalization constant’s impact on the model. We include an ablation below to demonstrate the impact of using unified normalization. The “e?” column indicates whether the model uses a learned normalization scalar or not. We also provide additional long-context benchmarks to further justify our approach.
>
> | Architecture | e? | Params/Tokens | State Sz. (MB/layer) | LM (Ppl. ↓) | FDA (Acc. ↑) | SWDE (Acc. ↑) | SQUAD (Acc. ↑) | RULER (Acc. ↑) | LM-Evals (Acc. ↑) |
> |---|---|---|---|---|---|---|---|---|---|
> | Hedgehog m = 1 | – | 1.2b/10b | 1.1 | 7.5 | 24.8 | 39.0 | 31.6 | 16.9 | 44.6 |
> | Hedgehog m = 2 | Y | 1.2b/10b | 2.1 | **7.4** | 26.2 | **39.6** | **32.5** | 20.8 | 44.5 |
> | Hedgehog m = 2 | N | 1.2b/10b | 2.1 | **7.4** | **29.6** | 39.5 | 31.6 | **32.0** | **44.8** |
> | Hedgehog | – | 1.2b/10b | 2.1 | 7.5 | 25.9 | 38.2 | 32.3 | 20.1 | 44.7 |
> | Hedgehog m = 4 | Y | 1.2b/10b | 4.2 | **7.4** | **26.4** | **45.4** | 31.6 | **29.0** | 44.2 |
> | Hedgehog m = 4 | N | 1.2b/10b | 4.2 | 7.5 | 22.1 | 40.7 | **34.0** | 24.4 | **45.5** |
> | Hedgehog | – | 1.2b/10b | 4.2 | 7.5 | 28.0 | 42.2 | 32.4 | 28.0 | 45.1 |
>
> Moreover, we would like to note that FDA, SWDE, and SQuAD are broadly used evaluation metrics for long-context recall and are considered to be strongly predictive of performance in a variety of settings and at scale. For example, the DeltaNet and Gated DeltaNet papers both evaluate their methods on these tasks. These methods have been successfully scaled-up in large models such as Qwen3-Next and Kimi Linear models, demonstrating that the trends found in these evaluations continue at scale.
>
> ## Intuition for learned feature maps.
>
> The original Hedgehog paper [1], Zhang et al discusses the impact of a learned linear projection in feature maps; their key finding is that it is highly effective in matching softmax quality. They provide extensive ablations of their use of a learned component and exp() activation, as well as comparisons to other popular feature maps. They demonstrate their approach’s efficacy by training 125M parameter models from scratch and distilling up to 7B parameter softmax attention models to using Hedgehog linear attention. Zhang et al also proposes using softmax in lieu of exp() activation to improve stability.
>
> To validate the stability of Hedgehog in our setting, we include additional stability metrics in our revised paper’s appendix. During training, we measure each parameter’s mean gradient value at regular step intervals. We find that gradient means for feature maps remain constant (i.e. the step-to-step difference is always near 0) across both feature map parameters, demonstrating that the training dynamics of our feature maps are stable.
>
> From a performance perspective, we point out that the parameter overhead of the learned feature maps is marginal, accounting for 0.09% of the overall parameter count in a model using 1 feature map (m=1) per linear attention layer.
>
> ## Path to softmax.
>
> Thank you for asking about this section of our paper, we have revised it to more clearly relate to the rest of the paper. The approximation argument presented in Figure 2 is meant to theoretically frame the relationship between softmax, polynomial approximation, and Hedgehog feature maps. In the Hedgehog paper, Zhang et al. seeks to construct a feature map that enables linear attention to mimic two key properties of softmax attention activations: spikiness and monotonicity. They identify the Taylor polynomial feature map to be a promising approach but bottlenecked by feature dimension size. Alternatively, Zhang et al proposes the Hedgehog feature map, which more efficiently approximates the spikiness and monotonicity properties of softmax via a learned component. This relationship between softmax, polynomial approximation, and learned feature maps motivates Cylon’s direction of scaling effective featurization width; a wider feature dimension implies a greater state size and more accurate approximation of softmax attention.
>
> In practice, we accumulate state in float32 for precision and stability.

---

> ### Author Response · Authors · 2025-11-23
>
> ## Quality Across Number of Partitions.
>
> Thank you for this question, it is helpful for us to clarify the presentation of our results. In Table 3 of our submission, we include figures demonstrating the end-to-end modeling quality of Cylon-partitioned models compared to baseline models. We demonstrate that increasing the number of feature maps for Cylon increases its performance on recall tasks, similar to how increasing the feature dimension would. Then, in Figure 3 we demonstrate that our kernel’s partitioning strategy runs faster at large state sizes compared to expanding the feature dimension in baseline kernels. To crystalize these trends, we have added end-to-end throughput metrics to Table 4 in our revised paper, as presented here.
>
> | Architecture | Params | State Sz. (MB/layer) | Prefill (Tok/Sec ↑) |
> |---|---|---|---|
> | Transformer | 1.3b | 33.5 | 9,656 |
> | Hedgehog m = 1 | 1.2b | 1.1 | 9,997 |
> | Hedgehog m = 2 | 1.2b | 2.1 | **9,670** |
> | Hedgehog | 1.2b | 2.1 | 8,449 |
> | Hedgehog m = 4 | 1.2b | 4.2 | **8,885** |
> | Hedgehog | 1.2b | 4.2 | 7,559 |
>
> To address your concerns about asynchronous overhead eventually outweighing the benefits of state remaining resident in registers, we refer to Figure 3 which demonstrates that Cylon scales more efficiently than other linear attention kernels. Cylon unlocks large state sizes (>= 131 MB) that other kernels encounter memory errors at. In our sweeps across numbers of partitions, we do not find that the asynchronous reduction ever outweighs the benefits of partitioning in comparison to baseline kernels. Since TMA runs asynchronously, this is largely attributable to the overlapping of TMA overheads with computation, as discussed in Section 3.3.
>
> ## Training Performance.
>
> We do not have a complete training kernel. However, gradient outputs across feature maps can be accumulated using the same asynchronous mechanisms that the forward pass utilizes (e.g. each SM computes independent gradients for shared parameters such as the QKV projection, which are then accumulated into global memory via TMA asynchronous reduction).
>
> ## Citations
>
> [1] Zhang, M., Bhatia, K., Kumbong, H., & Re, C. (2024). The Hedgehog & the Porcupine: Expressive Linear Attentions with Softmax Mimicry. The Twelfth International Conference on Learning Representations. https://openreview.net/forum?id=4g02l2N2Nx

---

### Official Review · Reviewer_mZng · 2025-11-01

**Soundness:** 3
**Presentation:** 4
**Contribution:** 3
**Rating:** 6
**Confidence:** 4

**Summary:**

This paper introduces CYLON,  a hardware-aware linear attention architecture, which partitions the recurrent state across the registers of multiple GPU processors and asynchronously combines the partitions. When applying CYLON linear attention to popular architectures, such as Hedgehog and Mamba-2,  it unlocks 3× higher throughput compared to prior linear attention algorithms for these architectures on both Hopper and Blackwell GPUs. Additionally, the proposed  CYLON makes large states available to model designers by unlocking
sizes (e.g., ≥ 131 MB) that are not by the existing linear attention kernels.

**Strengths:**

1. The paper introduces an asynchronous linear attention mechanism that allows decoupling of query and key-value updates. This improves efficiency and makes the model more flexible in streaming or autoregressive scenarios—an interesting departure from traditional synchronous designs.

2. The formulation appears mathematically grounded, with clear derivations showing how the asynchronous update maintains stability and approximate equivalence to standard attention under certain conditions.

3. The paper includes solid experimental validation on multiple benchmarks, showing that CYLON achieves better or comparable performance with less computation and memory, demonstrating both effectiveness and scalability.

**Weaknesses:**

1. Most experiments are on standard benchmarks. The benefits of asynchrony (e.g., in real-time or streaming settings) are not thoroughly demonstrated in practical large-scale or latency-critical applications
'
2. The intuition behind why asynchronous updates yield better representations is somewhat under-explained. The paper could clarify whether the asynchronous mechanism introduces a temporal inductive bias, improves stability, or reduces redundancy — and under what conditions this matters. It would be better to include visualizations (e.g., attention maps over time, update trajectories) or toy examples could help convey the underlying intuition.

3. While comparisons with standard linear attention methods are included, it may lack evaluation against the latest state-space models (like Mamba or Hyena), which could provide a stronger empirical contrast to justify the novelty and advantage.

**Questions:**

1. The paper claims that asynchronous updates between queries and key-values improve efficiency and representation quality. Could the authors elaborate on the mechanistic reason behind this improvement? For instance, does asynchrony introduce beneficial temporal diversity or implicit regularization, or does it merely approximate a form of delayed gradient update? Some empirical or theoretical clarification would help justify why asynchrony leads to better results beyond computational scheduling advantages.

2. Since state-space models (e.g., Mamba, Hyena) also enable efficient long-sequence modeling with sub-quadratic complexity, how does CYLON compare conceptually and practically to these approaches?

---

> ### Author Response · Authors · 2025-11-23
> **Response to Review**
>
> Thank you for your comments and questions! We have incorporated responses to your clarifying questions in the revised paper, which now provides clearer exposition of our methods and key contributions.
>
> ## Benefits of Async. in Downstream Applications
>
> In Section 1 and 2.2 of our paper, we discuss the motivation for scaling linear attention’s recurrent state size and the current bottlenecks to scaling state size in practice. Linear attention is an efficient alternative to traditional softmax attention. Its modeling quality is closely correlated with its recurrent state size, as is demonstrated by prior work like Zoology [1]. However, existing linear attention kernels fail to efficiently support large state sizes, as we discover via our profiling results in Section 2.3. Cylon addresses this barrier and enables linear attention algorithms to efficiently use larger state sizes.
>
> ## Benefits of Async. In Representations
>
> We believe there is a misunderstanding based on your response. Our approach does not introduce asynchronous updates between queries and key-values. Queries and key-values interact as they normally would in linear attention. Cylon distributes the computation of these interactions across GPU processors and uses asynchrony to overlap memory communication with computation. We refer to Section 3.1 of our paper for a detailed discussion of Cylon’s partitioning and how it is implemented on hardware.
>
> ## Comparisons to Baselines
>
> Regarding comparisons to other architectures, we clarify the relationship between Cylon and long-context mixers in our revised paper upload. In short, Cylon is not an architecture but rather a method for partitioning the recurrent state in any linear attention architecture. As displayed in Table 3, we apply Cylon to both Hedgehog linear attention and Mamba2, demonstrating that Cylon works with a variety of architectures.
>
> ## Citations
>
> [1] Arora, S., Eyuboglu, S., Timalsina, A., Johnson, I., Poli, M., Zou, J., Rudra, A., & Re, C. (2024). Zoology: Measuring and Improving Recall in Efficient Language Models. The Twelfth International Conference on Learning Representations. https://openreview.net/forum?id=LY3ukUANko

---

### Official Review · Reviewer_9j4n · 2025-11-01

**Soundness:** 3
**Presentation:** 3
**Contribution:** 2
**Rating:** 2
**Confidence:** 4

**Summary:**

Sequence models, crucial for tasks like information extraction and reasoning, face a trade-off between recall quality (ability to use long-sequence information) and memory efficiency. Linear attention models, in theory, can expand this trade-off frontier beyond other architectures like softmax attention and state-space models, but are limited by hardware bottlenecks, particularly regarding the size of their recurrent states. Current linear attention implementations struggle to scale beyond a few megabytes due to register memory exhaustion and expensive register spills, leading to significant slowdowns.

CYLON addresses this by partitioning the recurrent state across multiple GPU streaming multiprocessors (SMs) and asynchronously combining these partitions. This design leverages modern GPU features such as Tensor Memory Accelerators (TMA) for asynchronous I/O and reductions, and Tensor Cores for asynchronous multiplication. By ensuring each partition remains register-resident within an SM, CYLON maximizes locality and enables significantly larger effective state sizes. The paper demonstrates that this approach achieves up to 3x higher throughput compared to prior linear attention algorithms on both Hopper and Blackwell GPUs, and unlocks state sizes (e.g., ≥ 131 MB) that were previously inaccessible. CYLON-partitioned models maintain or exceed the quality of default linear attention models on language modeling and NLU tasks, effectively expanding the Pareto frontier for recall-efficiency.

**Strengths:**

1. The paper tackles a well-identified and significant challenge in linear attention: the inability to scale recurrent state sizes due to hardware memory limitations. This is a crucial problem for improving the recall quality of sequence models.
2. CYLON is meticulously designed with a deep understanding of modern GPU architectures, leveraging specific features like TMAs and Tensor Cores for asynchronous operations. This hardware-software co-design is key to its performance gains.
3. The reported 3x higher throughput and the ability to unlock much larger state sizes (≥ 131 MB) represent a substantial leap forward in the practical application of linear attention, particularly on modern GPUs.
4. Despite the architectural changes for efficiency, CYLON-partitioned models match or exceed the quality of non-partitioned linear attention models on various language modeling and NLU tasks, indicating that efficiency gains do not come at the cost of performance.
5. The paper includes theoretical proofs (Theorems 1, 2, 3) demonstrating that partitioning the state does not reduce the expressivity of linear attention and that the decomposition can be computationally efficient.

**Weaknesses:**

1. The asynchronous nature and distributed state management across SMs, while efficient, likely introduce significant complexity in implementation and debugging compared to traditional linear attention kernels.
2. While the paper discusses using learned MLPs for feature maps to ensure diversity, the specifics of how these distinct feature maps are learned and their impact on model training dynamics could be explored further.
3. The reliance on features like TMA and Warp-group matrix multiply means that CYLON's direct applicability might be limited to NVIDIA's Hopper and Blackwell architectures. While these are prevalent, performance on other hardware (e.g., AMD, custom AI accelerators) is not discussed.
4. The argument for "increasingly asynchronous machines" is based on current high-end NVIDIA GPUs. While this trend is likely to continue, future architectural shifts could potentially alter the optimal strategies for handling recurrent states.
5. While the paper evaluates language modeling and NLU tasks, a broader set of applications where long-context recall is paramount could further solidify CYLON's benefits.
6. While the paper mentions "long-latency roundtrips and inter-SM synchronization overhead," a more detailed breakdown of the specific overheads that CYLON successfully mitigates, perhaps with microbenchmark results for each component, could further strengthen the argument.

**Questions:**

see Weaknesses

---

> ### Author Response · Authors · 2025-11-23
> **Response to Review**
>
> Thank you for your suggestions and comments! We’ve included additional experimental results, profiling figures, and clarifications which we hope addresses your concerns and improves the strength of our work.
>
> ## Kernel Complexity
>
> Modifying existing H100 linear attention kernels to use Cylon-style partitioning takes very few modifications. In practice, we modify <10 lines of code from our baseline kernel. Other approaches to expanding state size, such as by increasing feature dimension, require more significant reimplementation. In our case, we find that modifying our baseline kernel to support a 2x larger feature dimension requires updating >30 lines of code.
>
> ## Learned Feature Map Training Dynamics
>
> The original Hedgehog paper [1], Zhang et al discusses the impact of a learned linear projection in feature maps; their key finding is that it is highly effective in matching softmax quality. They provide extensive ablations of their use of a learned component and exp() activation, as well as comparisons to other popular feature maps like Performer [2], 1 + ELU, ReLU, and cosFormer [3]. They demonstrate their approach’s efficacy by training 125M parameter models from scratch and distilling up to 7B parameter softmax attention models to using Hedgehog linear attention. To improve stability, Zhang et al proposes using softmax in lieu of an exp() activation. We adopt this approach.
>
> To validate the stability of Hedgehog in our setting, we include additional stability metrics in Section B.6 of our revised paper’s appendix. During training, we measure each parameter’s mean gradient value at regular step intervals. We find that gradient means for feature maps remain constant (i.e. the step-to-step difference is always near 0) across both feature map parameters, demonstrating that the training dynamics of our feature maps are stable.

---

> ### Author Response · Authors · 2025-11-23
>
> ## Additional Evaluations
>
> To increase the scope of our evaluations, we include (1) results from models trained on 5x the number of tokens (50Bn tokens), (2) results on a second pretraining corpus (the Pile, in addition to SlimPajama from our submission), and (3) results on additional long-context tasks from the popular RULER benchmark suite.
>
> ### Models trained on Slim Pajama
>
> | Architecture | Params/Tokens | State Sz. (MB/layer) | LM (Ppl. ↓) | FDA (Acc. ↑) | SWDE (Acc. ↑) | SQUAD (Acc. ↑) | RULER (Acc. ↑) | LM-Evals (Acc. ↑) |
> |---|---|---|---|---|---|---|---|---|
> | Transformer | 1.3b/10b | 33.5 | 12.2 | 73.1 | 72.3 | 24.4 | 41.7 | 46.0 |
> | Hedgehog m = 1 | 1.2b/10b | 1.1 | 12.4 | 22.3 | 40.3 | 33.3 | 21.2 | 45.7 |
> | Hedgehog m = 2 | 1.2b/10b | 2.1 | 12.3 | 25.2 | 46.8 | **35.9** | **26.2** | **46.6** |
> | Hedgehog | 1.2b/10b | 2.1 | 12.3 | **29.6** | **49.1** | 34.7 | 25.0 | 46.4 |
> | Hedgehog m = 4 | 1.2b/10b | 4.2 | 12.8 | **25.7** | 43.9 | 32.4 | 22.1 | **47.0** |
> | Hedgehog | 1.2b/10b | 4.2 | 12.7 | 22.9 | **44.7** | **32.7** | **23.4** | 45.8 |
> | Mamba-2 m = 1 | 1.3b/10b | 2.1 | 12.3 | 16.6 | 42.6 | 32.1 | 20.4 | 46.5 |
> | Mamba-2 m = 2 | 1.3b/10b | 4.2 | 12.1 | **23.8** | **45.2** | **34.6** | 26.4 | **46.6** |
> | Mamba-2 | 1.3b/10b | 4.2 | 12.7 | 16.7 | 41.3 | 30.0 | **29.5** | 46.5 |
> | Mamba-2 m = 4 | 1.3b/10b | 8.4 | 12.0 | **33.0** | **48.4** | 33.0 | **32.0** | 46.5 |
> | Mamba-2 | 1.3b/10b | 8.4 | 12.3 | 28.2 | 47.8 | **34.5** | 30.1 | **46.7** |
>
> ### Models trained on The Pile
>
> | Architecture | Params/Tokens | State Sz. (MB/layer) | LM (Ppl. ↓) | FDA (Acc. ↑) | SWDE (Acc. ↑) | SQUAD (Acc. ↑) | RULER Acc. ↑ | LM-Evals (Acc. ↑) |
> |---|---|---|---|---|---|---|---|---|
> | Transformer | 1.3b/10b | 33.5 | **7.2** | **76.0** | **68.6** | 36.5 | **46.3** | **45.2** |
> | Hedgehog m=1 | 1.2b/10b | 1.1 | 7.5 | 24.8 | 39.0 | 31.6 | 16.9 | 44.6 |
> | Hedgehog m=2 | 1.2b/10b | 2.1 | **7.4** | **26.2** | **39.6** | **32.5** | **20.8** | 44.5 |
> | Hedgehog | 1.2b/10b | 2.1 | 7.5 | 25.9 | 38.2 | 32.3 | 20.1 | **44.7** |
> | Hedgehog m=4 | 1.2b/10b | 4.2 | **7.4** | 26.4 | **45.4** | 31.6 | **29.0** | 44.2 |
> | Hedgehog | 1.2b/10b | 4.2 | 7.5 | **28.0** | 42.2 | **32.4** | 28.0 | **45.1** |
> | Mamba-2 m=1 | 1.3b/10b | 2.1 | 7.4 | 19.4 | 38.2 | 30.6 | 22.6 | 44.4 |
> | Mamba-2 m=2 | 1.3b/10b | 4.2 | 7.4 | 20.2 | 41.8 | **33.2** | 30.2 | **45.0** |
> | Mamba-2 | 1.3b/10b | 4.2 | **7.3** | **22.0** | **44.1** | 30.2 | **36.0** | **45.0** |
> | Mamba-2 m=4 | 1.3b/10b | 8.4 | **7.4** | **38.4** | **48.4** | 30.4 | 21.5 | **45.2** |
> | Mamba-2 | 1.3b/10b | 8.4 | **7.4** | 31.3 | 45.5 | **32.7** | **29.9** | 44.8 |
>
> ### Models trained on The Pile for 50B
>
> | Architecture | Params/Tokens | State Sz. (MB/layer) | LM (Ppl. ↓) | FDA (Acc. ↑) | SWDE (Acc. ↑) | SQUAD (Acc. ↑) | RULER (Acc. ↑) | LM-Evals (Acc. ↑) |
> |--------------|---------------|-------------------|----------------|----------------------------|------------------------------|-----------------|------------------------|------------------------|
> | Transformer | 1.3b/50b | 33.5 | 6.6 | 76.0 | 73.0 | 39.0 | 48.6 | 47.6 |
> | Hedgehog m=1 | 1.2b/50b | 1.1 | 6.9 | 21.23 | 44.73 | 35.19 | 19.97 | 46.34 |
> | Hedgehog m=2 | 1.2b/50b | 2.1 | **6.8** | 25.23 | **44.91** | 32.54 | 29.49 | 46.35 |
> | Hedgehog | 1.2b/50b | 2.1 | 6.9 | **29.85** | 43.65 | **34.22** | **34.32** | **46.79** |
> | Hedgehog m=4 | 1.2b/50b | 4.2 | **6.8** | **28.49** | 51.31 | **34.05** | **34.74** | 47.23 |
> | Hedgehog | 1.2b/50b | 4.2 | 6.9 | 27.77 | **54.37** | 33.11 | 34.68 | **47.32** |
>
> Please note that prior linear attention work such as GLA [4], DeltaNet [5], and Gated DeltaNet [6] evaluates 1.3B parameter models on FDA, SWDE, and SQuAD. We pick this parameter scale and set of tasks to follow literature standards and make our results as contextualizable as possible. Moreover, the recall-quality trends identified in the DetlaNet and Gated DeltaNet using these evals have scaled to larger models like Qwen3-Next and Kimi Linear, demonstrating that this evaluation setting is predictive of performance at scale in prior work.
>
> Regarding comparisons to other architectures, we clarify the relationship between Cylon and long-context mixers in our revised paper upload. In short, Cylon is not an architecture but rather a method for partitioning the recurrent state in any linear attention architecture. As displayed in Table 3, we apply Cylon to both Hedgehog linear attention and Mamba2, demonstrating that Cylon works with a variety of architectures.

---

> ### Author Response · Authors · 2025-11-23
>
> ## Broad Hardware Trends + Profiling
>
> As discussed in our general response, we find that both NVIDIA and AMD architectures are trending towards increasingly asynchronous operations. Moreover, in the paper we discuss other approaches to asynchronous reductions such as atomic add, which is supported by pre-Hopper hardware architectures and NVIDIA architectures as well. We hope the inclusion of these results demonstrates that the principles of asynchronous architecture design are more broad than a single hardware architecture or manufacturer.
>
> We also emphasize that TMA and async MMAs are not ephemeral hardware features, but rather are core to modern accelerators *across generations and hardware vendors*. Hopper, Blackwell, Grace Blackwell, and AMD’s gfx1250 hardware all offer TMA [7, 8, 9]. Hopper, Blackwell, and Grace Blackwell are the most widely used data center accelerators, so it is important to understand how to efficiently map AI architectures and algorithms to them.
>
> Thank you for your suggestion regarding a more detailed breakdown of memory overheads! In Section 3.3, we already include a discussion of reduction mechanisms and their comparative characteristics. To further highlight our claims, we also profile Cylon implemented with a fully-synchronous roundtrip global memory reduction which runs with up to 4x lower token throughput and over 10x lower memory throughput, motivating our transition from synchronous to asynchronous stores.
>
> | Mechanism | Feat. Maps | Latency (ms/iter) | Throughput (GB/s) | Notes |
> |-----------|------------|------------------|-------------------|--------|
> | Sync reductions | 4 | 5.3 | 82.6 | Barrier-synchronized global memory access |
> | | 8 | 15.8 | 28.2 | |
> | Atomic ops | 4 | 2.5 | 856.9 | Serialized global memory access |
> | | 16 | 9.3 | 845.4 | |
> | Async reductions | 4 | 2.2 | 932.3 | Overlaps reduction with compute |
> | | 16 | 7.9 | 940.9 | |
>
> ## Citations
>
> [1] Zhang, M., Bhatia, K., Kumbong, H., & Re, C. (2024). The Hedgehog & the Porcupine: Expressive Linear Attentions with Softmax Mimicry. The Twelfth International Conference on Learning Representations. https://openreview.net/forum?id=4g02l2N2Nx
>
> [2] Choromanski, K., Likhosherstov, V., Dohan, D., Song, X., Gane, A., Sarlos, T., Hawkins, P., Davis, J., Mohiuddin, A., Kaiser, L., Belanger, D., Colwell, L., & Weller, A. (2022). Rethinking Attention with Performers. https://arxiv.org/abs/2009.14794
>
> [3] Qin, Z., Sun, W., Deng, H., Li, D., Wei, Y., Lv, B., Yan, J., Kong, L., & Zhong, Y. (2022). cosFormer: Rethinking Softmax In Attention. International Conference on Learning Representations. https://openreview.net/forum?id=Bl8CQrx2Up4
>
> [4] Yang, S., Wang, B., Shen, Y., Panda, R., & Kim, Y. (2024). Gated Linear Attention Transformers with Hardware-Efficient Training. In R. Salakhutdinov, Z. Kolter, K. Heller, A. Weller, N. Oliver, J. Scarlett, & F. Berkenkamp (Eds.), Proceedings of the 41st International Conference on Machine Learning (Vol. 235, pp. 56501–56523). PMLR. https://proceedings.mlr.press/v235/yang24ab.html
>
> [5] Yang, S., Wang, B., Zhang, Y., Shen, Y., & Kim, Y. (2024). Parallelizing Linear Transformers with the Delta Rule over Sequence Length. In A. Globerson, L. Mackey, D. Belgrave, A. Fan, U. Paquet, J. Tomczak, & C. Zhang (Eds.), Advances in Neural Information Processing Systems (Vol. 37, pp. 115491–115522). Curran Associates, Inc. https://doi.org/10.52202/079017-3668
>
> [6] Yang, S., Kautz, J., & Hatamizadeh, A. (2025). Gated Delta Networks: Improving Mamba2 with Delta Rule. The Thirteenth International Conference on Learning Representations. https://openreview.net/forum?id=r8H7xhYPwz
>
> [7] https://resources.nvidia.com/en-us-hopper-architecture/nvidia-h100-tensor-c
>
> [8] https://developer.nvidia.com/blog/nvidia-blackwell-doubles-llm-training-performance-in-mlperf-training-v4-1/
>
> [9] https://github.com/triton-lang/triton/pull/8333, https://github.com/triton-lang/triton/pull/8392

---

### Official Review · Reviewer_cjQm · 2025-11-01

**Soundness:** 3
**Presentation:** 3
**Contribution:** 3
**Rating:** 6
**Confidence:** 4

**Summary:**

The paper tackles the practical limit that prevents linear-attention models from scaling recurrent state size due to GPU register pressure and spill-induced stalls. The proposed CYLON architecture partitions the linear-attention recurrent state across multiple streaming multiprocessors (SMs), keeps each partition register-resident, and aggregates partial results via asynchronous global reductions (TMA) while overlapping I/O and compute with wave specialization and async MMAs. Algorithmically, CYLON replaces a single feature map with per-partition maps and introduces a lightweight normalization relaxation with a learnable scalar. On Hopper and Blackwell GPUs, CYLON reports up to 3 times throughput improvements over widely used linear-attention kernels and enables large effective state sizes that prior kernels could not support. Language-model pretraining at 1.3B scale on 10B tokens shows that CYLON preserves perplexity and recall-oriented downstream performance relative to non-partitioned linear attention at matched state sizes. Theoretical results show that partitioning the feature map preserves expressivity and that simple polynomial expansions can be distributed across SMs.

**Strengths:**

- **(S1) Clear motivation grounded in profiling evidence.** The introduction and Section 2.3 convincingly identify register spills and long scoreboard stalls as the limiting factor for large linear‑attention states, supported by Nsight Compute profiles in Table 1. This isolates the true bottleneck and motivates the partition‑plus‑async design.

- **(S2) Hardware‑aware design with principled asynchrony.** CYLON combines async MMA, async I/O, and async global reductions to keep state in registers and avoid expensive HBM round trips. For example, Table 2 shows TMA reductions are 8–11% faster throughput and 12–15% lower latency than atomics in isolation, matching the end‑to‑end speedups. Table 3 reports comparable perplexity to non‑partitioned baselines at the same state size for both Hedgehog and Mamba‑2; recall‑heavy tasks improve as state size increases, with CYLON keeping pace or winning.

- **(S3) Reasonable reproducibility and clarity for a systems paper.** The manuscript is well organized and written in clear and professional English with comprehensive figures and tables. Meanwhile, the methodology is well structured and described with enough specificity (kernel pseudocode at the right abstraction, state‑size accounting, training stacks, hyper-parameters), and Appendices A–C document settings, extra sweeps, and proofs. Figures and writing are generally clear.

**Weaknesses:**

- **(W1) Limited theoretical depth relative to systems novelty.** The “equivalence” results largely formalize block decomposition of a unified feature map (Theorems 1–2) and a standard feature‑lifting observation for polynomials (Theorem 3). They do not analyze finite‑precision effects, async reduction order, or the learned scalar in Eq. (10) for stability/bias—precisely where implementation deviates from the idealized algebra. Strengthening this link would improve the scientific tightness.

- **(W2) Evaluation scope for model quality is narrow.** The main pre-training is 1.3B parameters on 10B tokens, with quality measured on standard NLU and three recall tasks. Long‑context tasks with explicit reasoning chains or multi‑hop retrieval are absent; comparisons to other efficient non‑LA long‑context mixers (e.g., SSM variants, RetNet) at comparable compute are not included. Hence, the claim that CYLON “expands the Pareto frontier” is well‑supported on throughput/state, but only suggestive on end‑task quality and reasoning.

- **(W3) Missing end‑to‑end inference metrics and profiling breakdowns.** Results emphasize iterations/sec for prefill at fixed (N, batch, d). However, decode‑token latency, HBM GB/s, occupancy, tensor‑core utilization, and power/energy are not reported, despite the contribution centering on memory hierarchy and asynchrony. Practitioners need these to make deployment decisions.

- **(W4) Minor issues of writing and limitations.** Firstly, the authors should double-check the typos in the main text.  Meanwhile, there are several limitations. (i) The effective partition count mm is bounded by available SMs and on‑chip memory, so speedups may vary substantially across SKUs (SXM vs PCIe) and future architectures with different async primitives. (ii) Overlapping TMA, async MMAs, and reductions increases kernel complexity and sensitivity to ISA/driver changes; without upstreaming, sustaining performance across generations may be non‑trivial.

**Questions:**

- **(Q1)** How are gradients and optimizer states synchronized with async reductions? Any measurable **gradient variance** or **load imbalance** across $\{\phi_k\}$ affecting convergence speed? Please report training stability signals (e.g., grad norms, step‑to‑step variance).

- **(Q2)** What are the end‑to‑end results (quality + stability) in FP8 for CYLON, given Figure 2’s approximation argument? What accumulation precisions are used along the reduction path, and are there guardrails (loss scaling, Kahan‑style compensation)?

- **(Q3)** Minor concerns about the fair experimental setup for decode latency. Were baseline kernels re‑tuned for large states (e.g., launch parameters, chunk shapes) before concluding the 3× speedup? Can you add decode latency (ms/token) at several state sizes to clarify real‑time implications?

---

> ### Author Response · Authors · 2025-11-23
> **Response to Review**
>
> Thank you for your thoughtful comments! We’ve included additional comments and data which we hope addresses your concerns and questions.
>
> ## W1. Theoretical Depth
>
> To mitigate effects from finite-precision and reduction order, we use float32 for state accumulation to avoid precision-issues. Regarding the learned scalar, although we do not make a theoretical claim regarding our approach, we include an additional ablation in our revised upload motivating our use of a learned normalization term. Moreover, prior work such as Mamba-v2 [1] and DeltaNet [2] entirely avoid normalization by opting for a pre-layer GroupNorm, LayerNorm, or RMSNorm. We train our models with RMSNorm prior to the Cylon layer.
>
> ## W2. Evaluation Scope
>
> To increase the scope of our evaluations, we include (1) results from models trained on 5x the number of tokens (50Bn tokens), (2) results on a second pretraining corpus (the Pile, in addition to SlimPajama from our submission), and (3) results on additional long-context tasks from the popular RULER benchmark suite. We present the average score of 12 tasks from RULER (the 13th task is SQUAD and is already presented in our original evaluation suite).
>
> ### Models trained on Slim Pajama
>
> | Architecture | Params/Tokens | State Sz. (MB/layer) | LM (Ppl. ↓) | FDA (Acc. ↑) | SWDE (Acc. ↑) | SQUAD (Acc. ↑) | RULER (Acc. ↑) | LM-Evals (Acc. ↑) |
> |---|---|---|---|---|---|---|---|---|
> | Transformer | 1.3b/10b | 33.5 | 12.2 | 73.1 | 72.3 | 24.4 | 41.7 | 46.0 |
> | Hedgehog m = 1 | 1.2b/10b | 1.1 | 12.4 | 22.3 | 40.3 | 33.3 | 21.2 | 45.7 |
> | Hedgehog m = 2 | 1.2b/10b | 2.1 | 12.3 | 25.2 | 46.8 | **35.9** | **26.2** | **46.6** |
> | Hedgehog | 1.2b/10b | 2.1 | 12.3 | **29.6** | **49.1** | 34.7 | 25.0 | 46.4 |
> | Hedgehog m = 4 | 1.2b/10b | 4.2 | 12.8 | **25.7** | 43.9 | 32.4 | 22.1 | **47.0** |
> | Hedgehog | 1.2b/10b | 4.2 | 12.7 | 22.9 | **44.7** | **32.7** | **23.4** | 45.8 |
> | Mamba-2 m = 1 | 1.3b/10b | 2.1 | 12.3 | 16.6 | 42.6 | 32.1 | 20.4 | 46.5 |
> | Mamba-2 m = 2 | 1.3b/10b | 4.2 | 12.1 | **23.8** | **45.2** | **34.6** | 26.4 | **46.6** |
> | Mamba-2 | 1.3b/10b | 4.2 | 12.7 | 16.7 | 41.3 | 30.0 | **29.5** | 46.5 |
> | Mamba-2 m = 4 | 1.3b/10b | 8.4 | 12.0 | **33.0** | **48.4** | 33.0 | **32.0** | 46.5 |
> | Mamba-2 | 1.3b/10b | 8.4 | 12.3 | 28.2 | 47.8 | **34.5** | 30.1 | **46.7** |
>
> ### Models trained on The Pile
>
> | Architecture | Params/Tokens | State Sz. (MB/layer) | LM (Ppl. ↓) | FDA (Acc. ↑) | SWDE (Acc. ↑) | SQUAD (Acc. ↑) | RULER Acc. ↑ | LM-Evals (Acc. ↑) |
> |---|---|---|---|---|---|---|---|---|
> | Transformer | 1.3b/10b | 33.5 | **7.2** | **76.0** | **68.6** | 36.5 | **46.3** | **45.2** |
> | Hedgehog m=1 | 1.2b/10b | 1.1 | 7.5 | 24.8 | 39.0 | 31.6 | 16.9 | 44.6 |
> | Hedgehog m=2 | 1.2b/10b | 2.1 | **7.4** | **26.2** | **39.6** | **32.5** | **20.8** | 44.5 |
> | Hedgehog | 1.2b/10b | 2.1 | 7.5 | 25.9 | 38.2 | 32.3 | 20.1 | **44.7** |
> | Hedgehog m=4 | 1.2b/10b | 4.2 | **7.4** | 26.4 | **45.4** | 31.6 | **29.0** | 44.2 |
> | Hedgehog | 1.2b/10b | 4.2 | 7.5 | **28.0** | 42.2 | **32.4** | 28.0 | **45.1** |
> | Mamba-2 m=1 | 1.3b/10b | 2.1 | 7.4 | 19.4 | 38.2 | 30.6 | 22.6 | 44.4 |
> | Mamba-2 m=2 | 1.3b/10b | 4.2 | 7.4 | 20.2 | 41.8 | **33.2** | 30.2 | **45.0** |
> | Mamba-2 | 1.3b/10b | 4.2 | **7.3** | **22.0** | **44.1** | 30.2 | **36.0** | **45.0** |
> | Mamba-2 m=4 | 1.3b/10b | 8.4 | **7.4** | **38.4** | **48.4** | 30.4 | 21.5 | **45.2** |
> | Mamba-2 | 1.3b/10b | 8.4 | **7.4** | 31.3 | 45.5 | **32.7** | **29.9** | 44.8 |
>
> ### Models trained on The Pile for 50B
>
> | Architecture | Params/Tokens | State Sz. (MB/layer) | LM (Ppl. ↓) | FDA (Acc. ↑) | SWDE (Acc. ↑) | SQUAD (Acc. ↑) | RULER (Acc. ↑) | LM-Evals (Acc. ↑) |
> |--------------|---------------|-------------------|----------------|----------------------------|------------------------------|-----------------|------------------------|------------------------|
> | Transformer | 1.3b/50b | 33.5 | 6.6 | 76.0 | 73.0 | 39.0 | 48.6 | 47.6 |
> | Hedgehog m=1 | 1.2b/50b | 1.1 | 6.9 | 21.23 | 44.73 | 35.19 | 19.97 | 46.34 |
> | Hedgehog m=2 | 1.2b/50b | 2.1 | **6.8** | 25.23 | **44.91** | 32.54 | 29.49 | 46.35 |
> | Hedgehog | 1.2b/50b | 2.1 | 6.9 | **29.85** | 43.65 | **34.22** | **34.32** | **46.79** |
> | Hedgehog m=4 | 1.2b/50b | 4.2 | **6.8** | **28.49** | 51.31 | **34.05** | **34.74** | 47.23 |
> | Hedgehog | 1.2b/50b | 4.2 | 6.9 | 27.77 | **54.37** | 33.11 | 34.68 | **47.32** |

---

> ### Author Response · Authors · 2025-11-23
>
> ## W2. Continued
>
> Please note that prior linear attention work such as GLA [3], DeltaNet [2], and Gated DeltaNet [4] evaluates 1.3B parameter models on FDA, SWDE, and SQuAD. We pick this parameter scale and set of tasks to follow literature standards and make our results as contextualizable as possible. Moreover, the recall-quality trends identified in the DetlaNet and Gated DeltaNet using these evals have scaled to larger models like Qwen3-Next and Kimi Linear, demonstrating that this evaluation setting is predictive of performance at scale in prior work.
>
> Regarding comparisons to other architectures, we clarify the relationship between Cylon and long-context mixers in our revised paper. In short, Cylon is not an architecture but rather a method for partitioning the recurrent state in any linear attention architecture. As shown in Table 3, we apply Cylon to both Hedgehog linear attention and Mamba2, demonstrating that Cylon works with a variety of architectures.
>
> ## W3. End-to-end Inference Metrics and Profiling
>
> We include the requested benchmarking numbers for HBM throughput, SM occupancy, tensor-core utilization, and power consumption. We compare our baseline linear attention kernel to Cylon across feature maps.
>
> | Kernel | Feature Maps | HBM Throughput (Gb/s) | Occupancy % | Tensor-Core Utilization | Energy (J) |
> |--------|--------------|----------------------|-------------|------------------------|------------|
> | Linear Attention | 1 | 893.95 | 12.46 | 29.44 | 1.37 |
> | Cylon | 1 | 834.47 | 12.36 | 24.04 | 1.41 |
> | Cylon | 2 | 919.29 | 12.36 | 24.09 | 2.83 |
> | Cylon | 4 | 926.48 | 12.36 | 24.13 | 5.63 |
> | Cylon | 8 | 932.30 | 12.36 | 24.15 | 11.56 |
> | Cylon | 16 | 940.88 | 12.36 | 24.16 | 23.74 |
>
> We note that we do not provide a decoding kernel.
>
> ## W4. Minor issues.
>
> Thank you for the feedback, we have carefully gone through the main text to look for any typos.
>
> Next we discuss the noted hardware limitations:
> - (i) **Effective partition count is bounded by the number of available SMs and available on-chip memory.** We note that the number of processors is rapidly increasing across GPU generations (132 processors on Hopper, 160 processors on Blackwell, 256 processors on AMD MI355X), and larger on-chip memory per processor (64KB on AMD MI300 to 160KB on AMD MI355) [5, 6, 7, 8, 9]. We note that the available memory on current chips already exceeds the KV-cache memory for a 3k token sequence from a 1.3B parameter Transformer.
> - (ii) **Overlapping TMA and async MMAs adds complexity and sensitivity across hardware generations.** We respectfully disagree that our use of these features is a limitation. We note that Cylon requires <10 lines of code to change between a vanilla linear attention and Cylon linear attention kernel. We also emphasize that TMA and async MMAs are not ephemeral hardware features, but rather are core to modern accelerators *across generations and hardware vendors*. Hopper, Blackwell, Grace Blackwell, and AMD’s gfx1250 hardware all offer TMA. Hopper, Blackwell, and Grace Blackwell are the most widely used data center accelerators, so it is important to understand how to efficiently map AI architectures and algorithms to them. Flash Attention 3 [10], COMET for MoE [11], ThunderKittens [12], TileLang [13], and many other systems rely on overlapping the use of these important hardware features.
>
> For completion, we provide additional benchmarks on an H100 SXM in Section B.1 of our revised paper’s appendix. Below is a tabular version of these results. We find that Cylon continues to achieve significantly faster token throughput compared to other linear attention kernels.
>
> | State Size (approx) | Kernel | Time (μs/iter) | Speedup vs Slowest |
> |---------------------|--------|----------------|-------------------|
> | **~1.0 MB** | | | |
> | | Cylon (m=1) | **354.34** | 3.9x |
> | | Chunk Linear (Head) | 997.12 | 1.4x |
> | | Chunk Mamba-2 (Feature Dim) | 1,398.45 | 1.0x |
> | **~2 MB** | | | |
> | | Cylon (m=2) | **695.31** | 3.4x |
> | | Chunk Linear (Head) | 1,403.08 | 1.7x |
> | | Chunk Linear (Feature Dim) | 1,755.96 | 1.3x |
> | | Chunk Mamba-2 (Feature Dim) | 2,365.55 | 1.0x |
> | **~4 MB** | | | |
> | | Cylon (m=4) | **1,359.34** | 3.5x |
> | | Chunk Linear (Head) | 2,509.41 | 1.9x |
> | | Chunk Linear (Feature Dim) | 3,221.50 | 1.5x |
> | | Chunk Mamba-2 (Feature Dim) | 4,769.44 | 1.0x |
> | **~8 MB** | | | |
> | | Cylon (m=8) | **2,692.62** | 3.3x |
> | | Chunk Linear (Head) | 4,853.19 | 1.8x |
> | | Chunk Linear (Feature Dim) | 6,148.88 | 1.5x |
> | | Chunk Mamba-2 (Feature Dim) | 8,974.65 | 1.0x |

---

> ### Author Response · Authors · 2025-11-23
>
> ## Q1. Training Details + Stability
>
> We do not have a complete training kernel. However, gradient outputs can also be accumulated using the same asynchronous mechanisms that the forward pass is implemented with (e.g. each SM computes independent gradients for the KVQ projection which are joined by summation).
>
> To validate training stability, we include additional stability metrics in Section B.6 of our revised paper’s appendix. During training, we measure each parameter’s mean gradient value at regular step intervals. We find that gradient means for feature maps remain constant (i.e. the step-to-step difference is always near 0) across both feature map parameters, demonstrating that the training dynamics of our feature maps are stable.
>
> ## Q2. Accumulation Precision
>
> Thank you for your questions regarding Figure 2 and the approximation argument. We have included additional context to our revised paper to clarify its relationship to our work. Figure 2 is meant to theoretically frame the relationship between softmax, polynomial approximation, and Hedgehog feature maps. In the Hedgehog paper, Zhang et al. seeks to construct a feature map that enables linear attention to mimic two key properties of softmax attention activations: spikiness and monotonicity [14]. They identify the Taylor polynomial feature map to be a promising approach but bottlenecked by feature dimension size. Alternatively, Zhang et al proposes the Hedgehog feature map, which more efficiently approximates the spikiness and monotonicity properties of softmax via a learned MLP with an exp() nonlinearity. Critically, they identify that the choice of exp() produces spikiness and monotonicity unlike other non-linearities like ReLU. Zhang et al demonstrates with an ablation that both the exp() and learned weights are necessary for achieving attention rivaling softmax in quality. This relationship between softmax, polynomial approximation, and learned feature maps motivates Cylon’s direction of scaling state size. By increasing the number of feature maps, we effectively compute a wider feature dimension which leads to a more accurate approximation of softmax attention.
>
> In practice, our models are trained with float32 accumulation for the state tensor for numerical stability purposes.
>
> ## Q3. Experimental Setup
>
> The baseline kernels are implemented in Triton, which performs extensive launch parameter autotuning. We benchmark all kernels after a number of “warmup” iterations such that effects from autotuning are not present and we report results using  the kernel’s best parameter choice. Moreover, each kernel is benchmarked with input tensor shapes selected from the kernel’s corresponding pretrained model. This is motivated by the authors choosing optimal input dimensions for their model’s performance. We benchmark kernels using these choices of hyperparameters such that they are presented as fairly as possible.

---

> ### Author Response · Authors · 2025-11-23
>
> ## Citations
>
> [1] Dao, T., & Gu, A. (2024). Transformers are SSMs: Generalized Models and Efficient Algorithms Through Structured State Space Duality. In R. Salakhutdinov, Z. Kolter, K. Heller, A. Weller, N. Oliver, J. Scarlett, & F. Berkenkamp (Eds.), Proceedings of the 41st International Conference on Machine Learning (Vol. 235, pp. 10041–10071). PMLR. https://proceedings.mlr.press/v235/dao24a.html
>
> [2] Yang, S., Wang, B., Zhang, Y., Shen, Y., & Kim, Y. (2024). Parallelizing Linear Transformers with the Delta Rule over Sequence Length. In A. Globerson, L. Mackey, D. Belgrave, A. Fan, U. Paquet, J. Tomczak, & C. Zhang (Eds.), Advances in Neural Information Processing Systems (Vol. 37, pp. 115491–115522). Curran Associates, Inc. https://doi.org/10.52202/079017-3668
>
> [3] Yang, S., Wang, B., Shen, Y., Panda, R., & Kim, Y. (2024). Gated Linear Attention Transformers with Hardware-Efficient Training. In R. Salakhutdinov, Z. Kolter, K. Heller, A. Weller, N. Oliver, J. Scarlett, & F. Berkenkamp (Eds.), Proceedings of the 41st International Conference on Machine Learning (Vol. 235, pp. 56501–56523). PMLR. https://proceedings.mlr.press/v235/yang24ab.html
>
> [4] Yang, S., Kautz, J., & Hatamizadeh, A. (2025). Gated Delta Networks: Improving Mamba2 with Delta Rule. The Thirteenth International Conference on Learning Representations. https://openreview.net/forum?id=r8H7xhYPwz
>
> [5] https://resources.nvidia.com/en-us-hopper-architecture/nvidia-h100-tensor-c
>
> [6] https://developer.nvidia.com/blog/inside-nvidia-blackwell-ultra-the-chip-powering-the-ai-factory-era/
>
> [7] https://www.amd.com/content/dam/amd/en/documents/instinct-tech-docs/product-briefs/amd-instinct-mi355x-gpu-brochure.pdf
>
> [8] https://www.amd.com/content/dam/amd/en/documents/instinct-tech-docs/white-papers/amd-cdna-3-white-paper.pdf
>
> [9] https://www.amd.com/content/dam/amd/en/documents/instinct-tech-docs/white-papers/amd-cdna-4-architecture-whitepaper.pdf
>
> [10] Shah, J., Bikshandi, G., Zhang, Y., Thakkar, V., Ramani, P., & Dao, T. (2024). FlashAttention-3: Fast and Accurate Attention with Asynchrony and Low-precision. The Thirty-Eighth Annual Conference on Neural Information Processing Systems. https://openreview.net/forum?id=tVConYid20
>
> [11] Zhang, S., Zheng, N., Lin, H., Jiang, Z., Bao, W., Jiang, C., Hou, Q., Cui, W., Zheng, S., Chang, L.-W., Chen, Q., & Liu, X. (2025). Comet: Fine-grained Computation-communication Overlapping for Mixture-of-Experts. https://arxiv.org/abs/2502.19811
>
> [12] Spector, B. F., Arora, S., Singhal, A., Parthasarathy, A., Fu, D. Y., & Re, C. (2025). ThunderKittens: Simple, Fast, and $\textit{Adorable}$ Kernels. The Thirteenth International Conference on Learning Representations. https://openreview.net/forum?id=0fJfVOSUra
>
> [13] Wang, L., Cheng, Y., Shi, Y., Tang, Z., Mo, Z., Xie, W., Ma, L., Xia, Y., Xue, J., Yang, F., & Yang, Z. (2025). TileLang: A Composable Tiled Programming Model for AI Systems. https://arxiv.org/abs/2504.17577
>
> [14] Zhang, M., Bhatia, K., Kumbong, H., & Re, C. (2024). The Hedgehog & the Porcupine: Expressive Linear Attentions with Softmax Mimicry. The Twelfth International Conference on Learning Representations. https://openreview.net/forum?id=4g02l2N2Nx

---

### Author Response · Authors · 2025-11-23
**Common Response to all Reviewers**

We thank all the reviewers for their time and effort reviewing our work and for their constructive comments, which have made our paper stronger. Reviewers consistently appreciated the “convincing,” “critical,” and “significant” characterization of register memory as the bottleneck in scaling linear attention quality [cjQm, b6Yf, 9j4n]. Reviewers agreed that our alleviation of this bottleneck “expands the Parteo frontier” of model design and represents a “substantial leap forward” in linear attention [b6Yf, 9j4n]. Compared to baselines, reviewers appreciated our evidence demonstrating that Cylon is “keeping pace or winning” on quality compared to baselines, thus demonstrating “effectiveness and scalability” [cjQm, 9j4n, mZng].

# Summary of Contributions

Our key contribution is a new hardware-aware strategy for using large amounts of memory (i.e., large recurrent states) with linear attention models. This is enabled by co-designing the algorithm with the  new hardware features introduced on modern data center GPUs (Hopper, Blackwell, AMD GFX1250) including dedicated asynchronous memory access hardware and asynchronous matrix multiply pipelines. This analysis is timely because there is a systematic shift across hardware vendors and generations towards these asynchronous features; the AI community has not fully studied how they impact the design of efficient architectures. Further, linear attention has been adopted at scale in models like Qwen3-Next, Minimax, Kimi Linear, and NVIDIA Nemotron, meaning that we need to understand how to make it hardware-efficient while preserving quality. In our work, we:

1. **Characterize GPU register memory as a key bottleneck in existing linear attention kernels which limits scaling downstream model quality.** Prior work demonstrates that linear attention quality scales closely with the size of its recurrent state [1, 2, 3, 4, 5]. However, prior linear attention kernels fail to efficiently support large recurrent state sizes since they exceed register memory capacity,

2. **Propose Cylon, a hardware-aware strategy for enabling linear attention models to  use large recurrent state sizes.** We demonstrate that our method is effective across different linear attention architectures (vanilla linear attention with Hedgehog feature maps and Mamba-v2). We also demonstrate its portability across modern datacenter hardware by providing kernels for both H100 and B200 GPUs.

3. **Evaluate the quality and speed of Cylon at large state sizes.** We demonstrate that Cylon can be applied to existing linear attention architectures, improving token throughput by up to 3x while preserving or improving model quality. We evaluate our models with general language models tasks (e.g. PiQA, ARC, etc.) and “recall” tasks, a class of evaluation which correlates closely with recurrent state size, as demonstrated in Based [2]. Cylon does not degrade recall accuracy, indicating that partitioning the recurrent state does not reduce its capacity.

---

> ### Author Response · Authors · 2025-11-23
>
> # Changes in Revision
>
> Overall, we include the following changes in this rebuttal response and our paper revision:
>
> 1. **Clarification of Cylon’s relationship to linear attention architectures.** Reviewers cjQm and mZng asked about the relationship between Cylon and linear attention architectures. We clarify that Cylon is a strategy for augmenting existing linear attention architectures, rather than a novel architecture. Our evaluations demonstrate that Cylon enables existing architectures to efficiently scale to larger state sizes.
>
> 2. **Clarifying the role of the learned feature maps.** Reviewers 9j4n and b6Yf asked for further explanation on the role of the learned feature maps that are used in Cylon’s approach. Both reviewers asked about how it impacts training stability. b6Yf also asked about the impact of additional complexity and parameter overhead. We analyze step-to-step gradient metrics from our training runs to demonstrate training stability. We include new discussion explaining how the feature maps help replicate the monotonicity and “spikiness” properties of the exp in attention softmax, and provide new ablations for the feature maps to support this explanation.
>
> 3. **Extended discussion of limitations.** As pointed out by reviewer cjQm, our partitioning approach can be bottlenecked by the available number of SMs (processors) on the GPU. We include discussion of this limitation to contextualize the trade-offs of our approach.
>
> 4. **Extended quality experiments.** Reviewers cjQm and 9j4n asked for additional long-context evaluations. We provide Cylon results for models trained on a second pre-training corpus (the Pile), trained for 5x more tokens (50Bn tokens), and evaluated on 12 new downstream tasks from the popular RULER benchmark suite. These trends continue to reinforce that Cylon matches the quality of existing linear attention architectures (e.g., Mamba-v2, Hedgehog), while running 3x more efficiently at large recurrent state sizes.
>
> 5. **Extended speed experiments.** Reviewer cjQm requested end-to-end inference metrics and profiles on H100 SXM. Reviewer b6Yf asked for clarification regarding the impact of the number of feature maps on token throughput. Reviewers cjQm and 94jn were also interested in additional profiling of Cylon and baseline kernels. We include additional experiments and profiles which demonstrate that Cylon improves end-to-end model throughput and achieves over 10x greater memory throughput compared to Cylon implemented with synchronous global memory reductions. We also show that Cylon’s token throughput improvement translates to H100 SXM.

---

> ### Author Response · Authors · 2025-11-23
>
> # Key New Experiments
>
> To further strengthen our claims surrounding state size scaling, we include additional results on new datasets, evals, and scaled up runs to further bolster our results.
>
> 1. **Extended evaluation scope.** In our original submission, we include results from models trained on Slim Pajama. In this response,  we include additional experiments from training on the Pile to demonstrate that the trends hold: Cylon preserves the quality of linear attention models (e.g., Mamba-v2, Hedgehog), while running up to 3x faster at large recurrent state sizes. We also train several Cylon models on 50B tokens to show that our trends continue as we scale the training duration. We further include results on 12 new tasks from the RULER [6] long-context task suite, reported as an average accuracy score. We find that Cylon continues to preserve or improve the accuracy of the baseline architectures.
>
> ### Models trained on Slim Pajama
>
> | Architecture | Params/Tokens | State Sz. (MB/layer) | LM (Ppl. ↓) | FDA (Acc. ↑) | SWDE (Acc. ↑) | SQUAD (Acc. ↑) | RULER (Acc. ↑) | LM-Evals (Acc. ↑) |
> |---|---|---|---|---|---|---|---|---|
> | Transformer | 1.3b/10b | 33.5 | 12.2 | 73.1 | 72.3 | 24.4 | 41.7 | 46.0 |
> | Hedgehog m = 1 | 1.2b/10b | 1.1 | 12.4 | 22.3 | 40.3 | 33.3 | 21.2 | 45.7 |
> | Hedgehog m = 2 | 1.2b/10b | 2.1 | 12.3 | 25.2 | 46.8 | **35.9** | **26.2** | **46.6** |
> | Hedgehog | 1.2b/10b | 2.1 | 12.3 | **29.6** | **49.1** | 34.7 | 25.0 | 46.4 |
> | Hedgehog m = 4 | 1.2b/10b | 4.2 | 12.8 | **25.7** | 43.9 | 32.4 | 22.1 | **47.0** |
> | Hedgehog | 1.2b/10b | 4.2 | 12.7 | 22.9 | **44.7** | **32.7** | **23.4** | 45.8 |
> | Mamba-2 m = 1 | 1.3b/10b | 2.1 | 12.3 | 16.6 | 42.6 | 32.1 | 20.4 | 46.5 |
> | Mamba-2 m = 2 | 1.3b/10b | 4.2 | 12.1 | **23.8** | **45.2** | **34.6** | 26.4 | **46.6** |
> | Mamba-2 | 1.3b/10b | 4.2 | 12.7 | 16.7 | 41.3 | 30.0 | **29.5** | 46.5 |
> | Mamba-2 m = 4 | 1.3b/10b | 8.4 | 12.0 | **33.0** | **48.4** | 33.0 | **32.0** | 46.5 |
> | Mamba-2 | 1.3b/10b | 8.4 | 12.3 | 28.2 | 47.8 | **34.5** | 30.1 | **46.7** |
>
> ### Models trained on The Pile
>
> | Architecture | Params/Tokens | State Sz. (MB/layer) | LM (Ppl. ↓) | FDA (Acc. ↑) | SWDE (Acc. ↑) | SQUAD (Acc. ↑) | RULER Acc. ↑ | LM-Evals (Acc. ↑) |
> |---|---|---|---|---|---|---|---|---|
> | Transformer | 1.3b/10b | 33.5 | **7.2** | **76.0** | **68.6** | 36.5 | **46.3** | **45.2** |
> | Hedgehog m=1 | 1.2b/10b | 1.1 | 7.5 | 24.8 | 39.0 | 31.6 | 16.9 | 44.6 |
> | Hedgehog m=2 | 1.2b/10b | 2.1 | **7.4** | **26.2** | **39.6** | **32.5** | **20.8** | 44.5 |
> | Hedgehog | 1.2b/10b | 2.1 | 7.5 | 25.9 | 38.2 | 32.3 | 20.1 | **44.7** |
> | Hedgehog m=4 | 1.2b/10b | 4.2 | **7.4** | 26.4 | **45.4** | 31.6 | **29.0** | 44.2 |
> | Hedgehog | 1.2b/10b | 4.2 | 7.5 | **28.0** | 42.2 | **32.4** | 28.0 | **45.1** |
> | Mamba-2 m=1 | 1.3b/10b | 2.1 | 7.4 | 19.4 | 38.2 | 30.6 | 22.6 | 44.4 |
> | Mamba-2 m=2 | 1.3b/10b | 4.2 | 7.4 | 20.2 | 41.8 | **33.2** | 30.2 | **45.0** |
> | Mamba-2 | 1.3b/10b | 4.2 | **7.3** | **22.0** | **44.1** | 30.2 | **36.0** | **45.0** |
> | Mamba-2 m=4 | 1.3b/10b | 8.4 | **7.4** | **38.4** | **48.4** | 30.4 | 21.5 | **45.2** |
> | Mamba-2 | 1.3b/10b | 8.4 | **7.4** | 31.3 | 45.5 | **32.7** | **29.9** | 44.8 |
>
> ### Models trained on The Pile for 50B
>
> | Architecture | Params/Tokens | State Sz. (MB/layer) | LM (Ppl. ↓) | FDA (Acc. ↑) | SWDE (Acc. ↑) | SQUAD (Acc. ↑) | RULER (Acc. ↑) | LM-Evals (Acc. ↑) |
> |--------------|---------------|-------------------|----------------|----------------------------|------------------------------|-----------------|------------------------|------------------------|
> | Transformer | 1.3b/50b | 33.5 | 6.6 | 76.0 | 73.0 | 39.0 | 48.6 | 47.6 |
> | Hedgehog m=1 | 1.2b/50b | 1.1 | 6.9 | 21.23 | 44.73 | 35.19 | 19.97 | 46.34 |
> | Hedgehog m=2 | 1.2b/50b | 2.1 | **6.8** | 25.23 | **44.91** | 32.54 | 29.49 | 46.35 |
> | Hedgehog | 1.2b/50b | 2.1 | 6.9 | **29.85** | 43.65 | **34.22** | **34.32** | **46.79** |
> | Hedgehog m=4 | 1.2b/50b | 4.2 | **6.8** | **28.49** | 51.31 | **34.05** | **34.74** | 47.23 |
> | Hedgehog | 1.2b/50b | 4.2 | 6.9 | 27.77 | **54.37** | 33.11 | 34.68 | **47.32** |

---

> ### Author Response · Authors · 2025-11-23
>
> 2. **Additional ablations and discussion of Cylon design choices.** To demonstrate the impact of our learned normalization term, we present ablations on this design choice. We also provide intuition for the learned feature maps along with references to prior work which validates learned featurization via extensive ablations.
>
> | Architecture | e? | Params/Tokens | State Sz. (MB/layer) | LM (Ppl. ↓) | FDA (Acc. ↑) | SWDE (Acc. ↑) | SQUAD (Acc. ↑) | RULER (Acc. ↑) | LM-Evals (Acc. ↑) |
> |---|---|---|---|---|---|---|---|---|---|
> | Hedgehog m = 1 | – | 1.2b/10b | 1.1 | 7.5 | 24.8 | 39.0 | 31.6 | 16.9 | 44.6 |
> | Hedgehog m = 2 | Y | 1.2b/10b | 2.1 | **7.4** | 26.2 | **39.6** | **32.5** | 20.8 | 44.5 |
> | Hedgehog m = 2 | N | 1.2b/10b | 2.1 | **7.4** | **29.6** | 39.5 | 31.6 | **32.0** | **44.8** |
> | Hedgehog | – | 1.2b/10b | 2.1 | 7.5 | 25.9 | 38.2 | 32.3 | 20.1 | 44.7 |
> | Hedgehog m = 4 | Y | 1.2b/10b | 4.2 | **7.4** | **26.4** | **45.4** | 31.6 | **29.0** | 44.2 |
> | Hedgehog m = 4 | N | 1.2b/10b | 4.2 | 7.5 | 22.1 | 40.7 | **34.0** | 24.4 | **45.5** |
> | Hedgehog | – | 1.2b/10b | 4.2 | 7.5 | 28.0 | 42.2 | 32.4 | 28.0 | 45.1 |
>
> 3. **Additional performance metrics and profiling.** We include (i) throughput for synchronous reductions, (ii) additional profiling metrics such as HBM throughput, (iii) kernel throughput numbers on H100 SXM, and (iv) end-to-end token throughput metrics.
>
> ### Synchronous reduction compared to atomic reductions and TMA async reduction
>
> | Mechanism | Feat. Maps | Latency (ms/iter) | Throughput (GB/s) | Notes |
> |-----------|------------|------------------|-------------------|--------|
> | Sync reductions | 4 | 5.3 | 82.6 | Barrier-synchronized global memory access |
> | | 8 | 15.8 | 28.2 | |
> | Atomic ops | 4 | 2.5 | 856.9 | Serialized global memory access |
> | | 16 | 9.3 | 845.4 | |
> | Async reductions | 4 | 2.2 | 932.3 | Overlaps reduction with compute |
> | | 16 | 7.9 | 940.9 | |
>
> ### Additional profiling metrics
>
> | Kernel | Feature Maps | HBM Throughput (Gb/s) | Occupancy % | Tensor-Core Utilization | Energy (J) |
> |--------|--------------|----------------------|-------------|------------------------|------------|
> | Linear Attention | 1 | 893.95 | 12.46 | 29.44 | 1.37 |
> | Cylon | 1 | 834.47 | 12.36 | 24.04 | 1.41 |
> | Cylon | 2 | 919.29 | 12.36 | 24.09 | 2.83 |
> | Cylon | 4 | 926.48 | 12.36 | 24.13 | 5.63 |
> | Cylon | 8 | 932.30 | 12.36 | 24.15 | 11.56 |
> | Cylon | 16 | 940.88 | 12.36 | 24.16 | 23.74 |
>
> ### Kernel throughput on H100 SXM
>
> | State Size (approx) | Kernel | Time (μs/iter) | Speedup vs Slowest |
> |---------------------|--------|----------------|-------------------|
> | **~1.0 MB** | | | |
> | | Cylon (m = 1) | **354.34** | 3.9x |
> | | Chunk Linear (Head) | 997.12 | 1.4x |
> | | Chunk Mamba-2 (Feature Dim) | 1,398.45 | 1.0x |
> | **~2 MB** | | | |
> | | Cylon (m = 2) | **695.31** | 3.4x |
> | | Chunk Linear (Head) | 1,403.08 | 1.7x |
> | | Chunk Linear (Feature Dim) | 1,755.96 | 1.3x |
> | | Chunk Mamba-2 (Feature Dim) | 2,365.55 | 1.0x |
> | **~4 MB** | | | |
> | | Cylon (m = 4) | **1,359.34** | 3.5x |
> | | Chunk Linear (Head) | 2,509.41 | 1.9x |
> | | Chunk Linear (Feature Dim) | 3,221.50 | 1.5x |
> | | Chunk Mamba-2 (Feature Dim) | 4,769.44 | 1.0x |
> | **~8 MB** | | | |
> | | Cylon (m = 8) | **2,692.62** | 3.3x |
> | | Chunk Linear (Head) | 4,853.19 | 1.8x |
> | | Chunk Linear (Feature Dim) | 6,148.88 | 1.5x |
> | | Chunk Mamba-2 (Feature Dim) | 8,974.65 | 1.0x |
>
> ### End-to-end token throughput
>
> | Architecture | Params | State Sz. (MB/layer) | Prefill (Tok/Sec ↑) |
> |---|---|---|---|
> | Transformer | 1.3b | 33.5 | 9,656 |
> | Hedgehog m = 1 | 1.2b | 1.1 | 9,997 |
> | Hedgehog m = 2 | 1.2b | 2.1 | **9,670** |
> | Hedgehog | 1.2b | 2.1 | 8,449 |
> | Hedgehog m = 4 | 1.2b | 4.2 | **8,885** |
> | Hedgehog | 1.2b | 4.2 | 7,559 |

---

> ### Author Response · Authors · 2025-11-23
>
> ## Citations
>
> [1] Arora, S., Eyuboglu, S., Timalsina, A., Johnson, I., Poli, M., Zou, J., Rudra, A., & Re, C. (2024). Zoology: Measuring and Improving Recall in Efficient Language Models. The Twelfth International Conference on Learning Representations. https://openreview.net/forum?id=LY3ukUANko
>
> [2] Arora, S., Eyuboglu, S., Zhang, M., Timalsina, A., Alberti, S., Zou, J., Rudra, A., & Re, C. (2024). Simple linear attention language models balance the recall-throughput tradeoff. ICLR 2024 Workshop on Mathematical and Empirical Understanding of Foundation Models. https://openreview.net/forum?id=qRlcoPhEoD
>
> [3] Bhattamishra, S., Hahn, M., Blunsom, P., & Kanade, V. (2024). Separations in the Representational Capabilities of Transformers and Recurrent Architectures. In A. Globerson, L. Mackey, D. Belgrave, A. Fan, U. Paquet, J. Tomczak, & C. Zhang (Eds.), Advances in Neural Information Processing Systems (Vol. 37, pp. 36002–36045). Curran Associates, Inc. https://doi.org/10.52202/079017-1135
>
> [4] Zubic, N., Soldà, F., Sulser, A., & Scaramuzza, D. (2025). Limits of Deep Learning: Sequence Modeling through the Lens of Complexity Theory. The Thirteenth International Conference on Learning Representations. https://openreview.net/forum?id=DhdqML3FdM
>
> [5] Jelassi, S., Brandfonbrener, D., Kakade, S. M., & Malach, E. (2024). Repeat After Me: Transformers are Better than State Space Models at Copying. https://arxiv.org/abs/2402.01032
>
> [6] Hsieh, C.-P., Sun, S., Kriman, S., Acharya, S., Rekesh, D., Jia, F., & Ginsburg, B. (2024). RULER: What’s the Real Context Size of Your Long-Context Language Models? First Conference on Language Modeling. https://openreview.net/forum?id=kIoBbc76Sy

---

### Meta-Review · Area_Chair_pLPy · 2026-01-06

**Summary:**

Reviewers broadly agreed that CYLON addresses a well-identified and practically important bottleneck in scaling linear attention models: the inability to increase recurrent state size due to GPU register pressure and register-spill–induced stalls. The hardware–software co-design, partitioning the recurrent state across SMs and overlapping computation with asynchronous memory operations, was widely viewed as insightful, timely, and impactful, with multiple reviewers noting that the work meaningfully expands the recall–efficiency Pareto frontier for linear attention. The concerns  focused on scope, clarity, and validation depth. After rebuttal, the paper demonstrates strong empirical validation, clear motivation, and careful systems-level analysis, with most reviewer concerns convincingly addressed. However, there are remaining issues primarily about theoretical depth and broader generalization (empirical validation beyond Hopper/Blackwell remains limited).

**Reviewer Concerns:**

**Concerns Addressed**

Expanded evaluation scope: The authors added extensive new experiments: training on an additional corpus (The Pile), longer training runs (50B tokens), and evaluations on 12 new long-context tasks from the RULER benchmark.

Detailed profiling and end-to-end metrics: The authors provided comprehensive profiling results, including HBM throughput, SM occupancy, tensor-core utilization, energy consumption, kernel-level timing on H100 SXM, and end-to-end token throughput.

Clarification of algorithmic design: The rebuttal clarified that CYLON is not a new attention architecture, but a hardware-aware strategy applicable to existing linear attention models (e.g., Hedgehog, Mamba-2).

Ablations on key design choices: New ablations addressed the learned normalization relaxation, feature-map design, and number of partitions, showing stable training dynamics and minimal parameter overhead.

Limitations and hardware discussion: The revised manuscript explicitly discusses limitations related to SM count, hardware variability, and kernel complexity, contextualizing the trade-offs and addressing requests for a clearer limitations section.

**Still Outstanding**

Depth of theoretical analysis: While reviewers largely accepted the empirical positioning, some concerns remain that the theory does not fully characterize finite-precision effects, async reduction order, or long-term stability.

Generality beyond modern datacenter GPUs: Although the authors argue convincingly that asynchronous features are a broad hardware trend (including AMD), empirical validation beyond Hopper/Blackwell remains limited.

**Reviewer Scores:**

Reviewer cjQm: Initial 6. Given that their major concerns (evaluation breadth, profiling, limitations) were directly addressed with substantial new experiments and metrics, but "depth of theoretical analysis" was not addressed, likely keep 6.

Reviewer b6Yf: Initial 4. After added ablations on normalization, feature-map complexity, profiling, and extended evaluations, their core concerns appear addressed. Likely increase to 6.

Reviewer 9j4n: Initial 2. Given the extensive clarifications, expanded experiments, but "empirical validation beyond Hopper/Blackwell remains limited", likely increase to 4.

Reviewer mZng: Initial 6. Their primary concern stemmed from a misunderstanding of asynchrony, which was explicitly clarified.Likely remain 6.

---

### Decision · Program_Chairs · 2026-01-26

Reject